# Eddy covariance fluxes of $CO_2$, $CH_4$ and $N_2O$ on a drained peatland forest after clearcutting

Olli-Pekka Tikkasalo[1], Olli Peltola[1], Pavel Alekseychik[1+], Juha Heikkinen[1+], Samuli Launiainen[1+], Aleksi Lehtonen[1+], Qian Li[1+], Eduardo Martínez-García[1+], Mikko Peltoniemi[1+], Petri Salovaara[1+], Ville Tuominen[2+] and Raisa Mäkipää[1]

[1]Natural Resources Institute Finland (Luke), 00791 Helsinki, Finland
[2]Finnish Meteorological Institute (FMI), 00560 Helsinki, Finland

[+] Authors are listed in an alphabetical order

*Correspondence to*: Olli-Pekka Tikkasalo, olli-pekka.tikkasalo@luke.fi

## Abstract

Rotation forestry based on clearcut harvesting, site preparation, planting, and intermediate thinnings is currently the dominant management approach in Fennoscandia. However, understanding of the greenhouse gas (GHG) emissions following clearcutting remains limited, particularly on drained peatland forests. In this study, we report eddy covariance-based (EC) net emissions of carbon dioxide ($CO_2$), methane ($CH_4$) and nitrous oxide ($N_2O$) from a boreal fertile drained peatland forest one year after the harvest. Our results show that on annual scale, the site was a net $CO_2$ source. The $CO_2$ emissions dominate the total annual GHG balance ( $23.3\,t\,CO_2$-eq ha$^{-1}$yr$^{-1}$ [ $22.4-24.1\,t\,CO_2$-eq ha$^{-1}$yr$^{-1}$ depending on the EC gap-filling method], 82.0% of the total), while the role of $N_2O$ emissions ($5.0\,t\,CO_2$-eq ha$^{-1}$yr$^{-1}$ [$4.9-5.1\,t\,CO_2$-eq ha$^{-1}$yr$^{-1}$], 17.6%) was also significant. The site was a weak $CH_4$ source ($0.1\,t\,CO_2$-eq ha$^{-1}$yr$^{-1}$ [$0.1-0.1\,t\,CO_2$-eq ha$^{-1}$yr$^{-1}$], 0.4%). A statistical model was developed to estimate surface-type-specific $CH_4$ and $N_2O$ emissions. The model was based on air temperature, soil moisture and contribution of specific surface-types within the EC flux footprint. The surface-types were classified using unmanned aerial vehicle (UAV) spectral imaging and machine learning. Based on the statistical models, the highest surface-type specific $CH_4$ emissions occurred from plant-covered ditches and exposed peat, while the surfaces dominated by living trees, dead wood, litter and exposed peat were the main contributors to $N_2O$ emissions. Our study provides new insights into how $CH_4$ and $N_2O$ fluxes are affected by surface-type variation across clearcutting areas in boreal forested peatlands. Our findings highlight the need for integrating surface-type-specific flux modelling, EC-based data, and chamber-based flux measurements to comprehend the GHG emissions following clearcutting and regeneration. Results strengthen the accumulated evidence that recently clearcut peatland forests are significant GHG sources.

# 1 Introduction

Globally, peatland soils store 650000 Mt of carbon (C), which is equivalent to more than half of the C in the atmosphere (FAO, 2020). In Europe, the estimated peatland C stock is 43620 Mt C, with a total peatland area of 58.8 Mha of which 46% is drained (UNEP, 2022). Drainage lowers water table depth (WTD) and accelerates aerobic peat decomposition, resulting in carbon dioxide ($CO_2$) emissions and an annual loss of soil C stock equivalent to 160 Mt C (UNEP, 2022). In the specific context of Finland, the greenhouse gas (GHG) flux balance in forested peatlands has been quantified at both the stand-level (Korkiakoski et al., 2023; Mäkiranta et al., 2010; Ojanen et al., 2010)  and the national scale (Alm et al., 2023; Statistics Finland, 2022). However, the short-term impact of clearcutting and following site preparation on the GHG fluxes of forested peatlands remains unclear and is not currently included in the national GHG inventories. Therefore, estimates of the current GHG balance of forested peatlands under management are associated with a considerable degree of uncertainty.

Rotation forestry is currently the dominant forest management method in Fennoscandia,. It is characterized by forest stands with an even age structure, resulting from forest regeneration by clearcutting and later by intermediate thinnings from below (Kuuluvainen et al., 2012). In Finland, 4.7 Mha of peatlands have been drained for forestry purposes (Korhonen et al., 2021). A large fraction of fertile drained peatland forests is currently at mature stage and approaching the decision of final harvesting and regeneration (Lehtonen et al., 2023). In rotation-based peatland forestry, clearcutting typically leads to maintenance ditching to ensure adequate drainage for undisturbed tree growth (Päivänen and Hånell, 2012). However, rotation forestry that involves clearcutting and maintenance ditching has been found to have several short-term negative external effects (Nieminen et al., 2018). These include increases in nutrient and dissolved organic carbon (DOC) exports to watercourses (Palviainen et al., 2022), loss of biodiversity (Paillet et al., 2010; Rajakallio et al., 2021), and enhanced $CO_2$ emissions (Korkiakoski et al., 2023). The magnitude and duration of the major GHG emissions – $CO_2$, methane ($CH_4$) and nitrous oxide ($N_2O$) – on boreal drained forested peatlands after clearcutting remain largely unclear. This is because there have been only a few studies assessing them to date (Korkiakoski et al., 2019, 2023; Mäkiranta et al., 2010; Tong et al., 2022). The lack of information on how clearcutting affects GHG emissions in boreal forestry-drained peatlands prevents the comparisons of climate change impacts of business-as-usual forestry (i.e., rotation) and alternative forest management methods (e.g., continuous cover forestry) (Kaarakka et al., 2021; Mäkipää et al., 2023).

Tree removal alters the local microclimate of forested peatlands by changing e.g., the amount of radiation available on the ground (Tikkasalo et al., 2024). This can result in higher soil temperatures (Pumpanen et al., 2004; Wu et al., 2011), potentially increasing peat decomposition and $CO_2$ emission rates (Jandl et al., 2007). On the other hand, piles of harvest residues may decrease the soil temperature creating biotic and abiotic variation. Under drained or unsaturated moisture conditions, this process may be further enhanced due to increased oxygen availability in soil (Maljanen et al., 2010; Ojanen et al., 2013; Drzymulska, 2016). The harvest of trees in peatland forests raise the WTD by decreasing transpiration and interception

(Sarkkola et al., 2010; Leppä et al., 2020a, b). This, in turn, may result in a slower peat decomposition rate. Furthermore, the removal of trees and decline of forest-floor vegetation will lead to a strong immediate reduction in photosynthesis in clearcutting areas. Drainage can increase root aeration and nutrient availability, which may benefit the rapid establishment of initial forest-floor vegetation and tree seedlings (Mäkiranta et al. 2010) and enhances rates of ground vegetation carbon sequestration (Minkkinen et al., 2001). However, ground vegetation is insufficient to compensate for the increase in ecosystem respiration caused by the decomposition of logging residues (Mäkiranta et al., 2012; Ojanen et al., 2017; Korkiakoski et al., 2019; Tong et al., 2022). Consequently, clearcutting transforms forested peatland ecosystems into net $CO_2$ sources during the early stages of stand development (Mäkiranta et al., 2010; Tong et al., 2022; Korkiakoski et al., 2023).

Peatland drainage has decreased $CH_4$ emissions compared to pristine peatlands, due to improved soil aeration (Maljanen et al., 2010; Ojanen et al., 2010). After tree-removal WTD typically rises (Korkiakoski et al., 2019; Leppä et al., 2020a), which supports the production of $CH_4$ in the extended anaerobic zone. This can turn peatland sites from net $CH_4$ sinks into sources (Korkiakoski et al., 2019). However, Ojanen et al. (2010, 2013) found that $CH_4$ emissions only increase when the WTD is at shallow level (i.e., within 30 cm from the soil surface). Furthermore, the response of vegetation to drainage may affect the supply of substrate to methanogens (Minkkinen and Laine, 2006), which can further enhance or offset the hydrological effects of drainage on $CH_4$ fluxes.

Clearcutting not only affects C fluxes, but also leads to increased $N_2O$ emissions (Robertson et al., 1987; Huttunen et al., 2003; Saari et al., 2009; Neill et al., 2006; Korkiakoski et al., 2019). This is due to the flush of decomposing logging residues and reduced nitrogen (N) uptake due to lower plant biomass, which both increase available soil N in the first years after the harvesting (Mäkiranta et al., 2012). $N_2O$ production is also favoured by redox conditions that vary between oxidative and reductive, which exist in wet but unsaturated peat after clearcutting and drainage. The production of $N_2O$ responds to changes in soil moisture, so the effect of drainage on $N_2O$ emissions is likely to depend on the combination of WTD change and soil nutrient status (Tong et al., 2022). Additionally, drying-rewetting events occurring during the growing season have been identified as 'hot moments' for $N_2O$ emissions (Groffman et al., 2009). Nevertheless, the accurate estimation of $N_2O$ emissions has remained a significant challenge due to their considerable spatio-temporal variations (Rautakoski et al., 2024), which are a consequence of the inherent complexity of the various interacting processes. Furthermore, given that $N_2O$ is a potent long-lived GHG and a stratospheric ozone-depleting substance that has been accumulating rapidly in the atmosphere over the last decades (Tian et al., 2024), it is important to determine the role of clearcutting in regulating the global GHG budget.

Most studies on GHG fluxes in boreal drained forested peatlands after clearcutting are based on manual chamber measurements (e.g., Mäkiranta et al., 2010; Tong et al., 2022). However, the magnitude and controls on $CO_2$, $CH_4$, and $N_2O$ fluxes in these high-latitude northern ecosystems remain highly uncertain. This is mainly related to the poor spatial and temporal representation of manual chamber-based GHG measurements (Savage and Davidson, 2003). Clearcutting creates a highly

heterogeneous surface, which makes it challenging to interpret ecosystem GHG fluxes due to variation in surface-specific fluxes. Previous research has demonstrated that forest-floor vegetation heterogeneity, logging residues, and ditches cause significant spatial variability in GHG fluxes from drained peatlands and clearcut areas (Minkkinen and Laine, 2006; Ojanen et al., 2010; Mäkiranta et al., 2012; Rissanen et al., 2023). In this context, eddy covariance (EC) has become a widely used technique for measuring the GHG exchange (Baldocchi, 2003) due to its ability to provide high-temporal resolution exchange rates integrated over a relatively large area. The EC footprint (i.e. source area of the measured flux) collects the contributions of each element of the surface area to the measured vertical turbulent flux (Vesala et al., 2008). Therefore, this area could be divided into distinct surface-types that form a heterogeneous matrix, enabling direct assessments of each surface-type on the measured GHG fluxes. While studies attributing EC measured surface fluxes to specific surface-types at heterogeneous ecosystems exist (Forbrich et al., 2011; Franz et al., 2016; Tuovinen et al., 2019; Ludwig et al., 2024), none of them focus on heterogeneous clearcut areas. The likely reason for this is the lack of high-resolution data on surface-types within the EC tower's footprint. The use of high-resolution georeferenced imagery from unmanned aerial vehicle (UAV) surveys, and the possibility to derive detailed surface maps, however, now enables the integration of footprint models and GHG flux measurements and attributing measured fluxes to specific surface features.

In light of the preceding considerations, there is a considerable degree of uncertainty associated with the magnitude of GHGs as well as their key modulating processes and spatial and temporal heterogeneity on boreal drained peatland forests under forestry management. This deficiency is directly attributable to the paucity of available studies on the subject. It is therefore imperative to improve our understanding on the impact of clearcutting and different surface types on GHG fluxes. Here, we examined the $CO_2$, $CH_4$, and $N_2O$ fluxes from a fertile boreal drained peatland forest located in southern Finland during the first full year (second growing season) after clearcutting. GHG fluxes were measured using an EC system during the year 2022, while clear-cutting was conducted during the winter and spring 2021. Information on surface-type variation across the footprint area was collected through drone imaging in June 2022. Our specific aims were to:

1. Quantify the magnitude and temporal variation of $CO_2$, $CH_4$, and $N_2O$ fluxes along their annual balances.
2. Estimate the differences in surface-type specific $CH_4$ and $N_2O$ fluxes, as well as their sensitivity to environmental variation.

## 2 Materials and methods

### 2.1 Measurement site

Ränskälänkorpi study site is a boreal peatland forest (ca. 24 ha) located in Southern Finland (61°11'N, 25°16'E, 144 m a.s.l.; Fig. 1, Fig. S1), which has been drained for forestry before 1960's. The climate is humid continental with a 30-year (1981–2022) mean annual air temperature and precipitation sum of 4.2°C and 611 mm, respectively. Air temperature and precipitation were obtained from a 10 km × 10 km grid of daily weather data from the Finnish Meteorological Institute (FMI)

resulting from a kriging-interpolation procedure (Venäläinen and Heikinheimo, 2002). The site maintains snow cover on average for 133 days, typically from early November to late April. The forest is dominated by Norway spruce (*Picea abies* (L.) Karst., about 70% of all trees), with some Scots pine (*Pinus sylvestris* L.) and Downy birch (*Betula pubescens* Ehrh.). The forest-floor vegetation is sparse and consists of mosses (mainly *Hylocomium splendens, Pleurozium schreberi* and *Dicranum polysetum*), dwarf shrubs (mainly *Vaccinium myrtillus* and *Vaccinium vitis-idaea*), as well as forbs such as *Dryopteris carthusiana*, *Gymnocarpium dryopteris*, *Trientalis europaea*, and *Oxalis acetosella*. The site consists of sedge-wood dominated peat, which is mainly more than 1 m deep. The site is a fertile and well-drained, Norway spruce dominated and represents mainly nutrient-rich Herb-rich (Rhtkg II) and *Vaccinium myrtillus* (Mtkg II) site types according to Finnish site types of drained peatland forests (Laine et al., 2012). In March 2021, the site was divided into three areas with different harvest treatments: non-harvested control (ca. 7.3 ha), selection harvest (CCF, ca. 10.0 ha), and clearcutting (CC, ca. 6.1 ha). The harvesting in the CCF and CC areas took place with harvester machinery primarily from 18th March to 1st April 2021 when the soil was frozen. The harvesting was completed in June 2021 in the north-western section of the CC area. This study was conducted in the CC area, where all the trees were cut. Some large, dead trees were retained on site, and the resulting logging residues (i.e., foliage, branches and stumps) were left on the ground. The understory vegetation was significantly impacted by the disturbance caused by the harvester and logging machines. The mean peat soil C/N ratio across 0-20 cm depth at the CC area was 32.1 ± 4.7 (C% = 54.8 ± 1.5 and N% = 1.7 ± 0.2; $n$ = 6 sampling locations). The WTD relative to the peat surface during the 2022 growing season ranged from –9 to –64 cm with an average value of –28 ± 9 cm ($n$ = 8 sampling locations). The stand regeneration was carried out in summer 2021 through ditch mounding and planting of Norway spruce seedlings, with an approximate density of $1800 - 2000$ seedlings $ha^{-1}$. The harvest and regeneration are according to common practices for operational forestry in Finland.

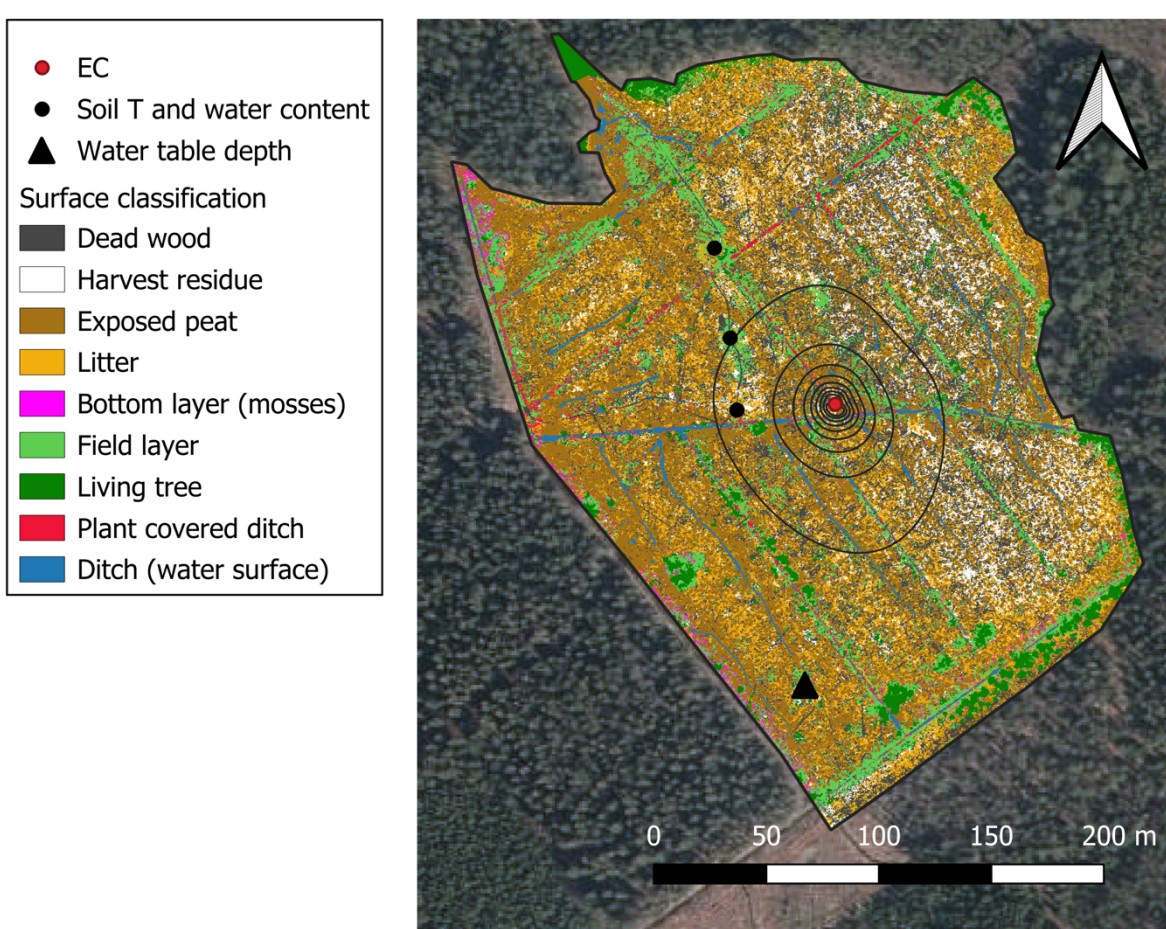

**Figure 1. Surface-type classification and aerial view of the experimental setup in the clearcut area.** Black triangle shows the location of water table depth measurement, black circles show the location of the soil temperature and moisture sensors, red circle shows the location of the eddy covariance (EC) tower. The contour lines display the mean footprint area (10 to 90[th] percentiles) for the year 2022. The pixel colour indicates the surface-type. The background aerial photo is acquired from the National Land Survey of Finland Topographic Database (distributed with CC-BY 4.0 license, retrieved 06/2024).

## 2.2 EC measurements

Ecosystem-atmosphere greenhouse gas exchange was measured with the EC technique in the middle of the CC area with a 3.1 m tall tower (see Fig. 1). Distance from the tower to the forest edge was at minimum 100 m in all directions. High frequency data on the three wind components and sonic temperature were acquired with an ultrasonic anemometer (uSonic-3 Cage MP, METEK GmbH, Germany), $CO_2$ and water vapor ($H_2O$) mixing ratios with a nondispersive infrared sensor (LI-7200RS, LI-COR Biosciences, NE, USA) and $CH_4$ and $N_2O$ mixing ratios with Tunable Infrared Laser Direct Absorption Spectrometer

(TILDAS, Aerodyne Research Inc, USA). All the EC data were logged with 10 Hz frequency. TILDAS data were logged to separate files and combined with the other EC data during data post-processing. TILDAS was located in a small, air-conditioned measurement hut and it was sampling the air with a 9 m long heated Teflon tube. Rapid flow in the tube was created with a scroll pump (TriScroll 600, Agilent Technologies Inc, USA). LI-7200RS was situated in the measurement tower and it was sampling the air with a heated sampling tube distributed with the instrument (ca. 0.7 m long tube with 5.3 mm inner diameter) and pump. The gas analysers sampling inlets were located next to the sonic anemometer (0.18 m horizontal separation).

In addition to the EC fluxes, several environmental variables were continuously monitored at the EC station. These include photosynthetically active radiation (PAR; LI-190R Quantum Sensor, LI-COR Biosciences, USA), air temperature ($T_{air}$) and humidity (HMP155 Humidity and Temperature Probe, Vaisala Oyj, Finland), shortwave and longwave incoming and outgoing radiation component (CNR4 4-component Net Radiometer, Kipp & Zonen, the Netherlands), precipitation (P; TR-525M Rainfall sensor, Texas Electronics, USA), soil temperature ($T_{soil}$) and water content ($\theta$) at 10 cm depth (Hydra Probe II, Stevens Water Monitoring Systems Inc., USA). These variables were logged with 1 min time step. Soil temperature and water were monitored also at other locations at the clearcut (see Fig. 1) with TMS-4 microclimate loggers (Standard datalogger, TOMST s.r.o, Prague, Czechia) and water table depth were measured with Odyssey Capacitance Water Level Logger (Dataflow Systems Ltd, New Zealand).

## 2.3 EC data processing

EC flux data processing followed international standards set e.g. by Integrated Carbon Observation System (ICOS) network (Franz et al., 2018) as much as feasible. Flux calculations were executed with the EddyPro open-source software (version 7.0.7, LI-COR Inc, USA). Fluxes were calculated using 30-min averaging time and turbulent fluctuations were separated from the measurements using block-averaging. The high frequency time series were despiked following Mauder et al (2013). High frequency gas data were already converted to dry mixing ratios internally by the measurement devices (LI-7200RS and TILDAS) and hence no conversions were done during post-processing. The gas sampling system (tubes and filters) induced time lags between wind and gas mixing ratio data. These time lags were estimated using cross-covariance maximisation and accounted for before flux calculations. Also, the flow coordinates were rotated using sector-wise planar fitting (Rannik et al., 2020) before calculating the covariances (i.e., fluxes) between the vertical wind component and gas mixing ratio time series. EC fluxes are always underestimated due to high frequency and low frequency dampening of the signal caused by the measurement system (e.g., dampening of the gas fluctuations in the sampling lines) and the need to use a finite flux averaging time, respectively. This underestimation of gas fluxes was corrected in this study following the approach by Fratini et al. (2012) and Moncrieff et al. (2005) with the exception that the cut-off frequencies characterising the high frequency dampening of each gas signal were estimated based on cospectra between vertical wind and gas time series and not from gas power spectra following Peltola et al. (2021).

The fully processed gas flux time series resulting from the processing procedure described above were quality filtered following Vitale et al. (2020) with few differences. First, flux data were discarded if the flux values were outside predefined limits, instrument diagnostics signalled erroneous measurement or site diaries suggested disturbance to the data. Then, the procedure by Vitale et al. (2020) was followed with the exception that the statistical model used in the quality filtering procedure was estimated using singular spectrum analysis and low-rank reconstruction of the time series (Golyandina et al., 2001; Mahecha et al., 2007) instead of the multiplicative model used in Vitale et al. (2020). After quality filtering, low turbulence periods during which EC fluxes do not represent surface-atmosphere exchange were identified using friction velocity and periods when friction velocity was below a site-specific threshold ( $0.09$ m s$^{-1}$) were removed from further analysis. After this procedure the flux data coverages were 64 %, 57 % and 57 % for $CO_2$, $CH_4$ and $N_2O$ flux time series, respectively, with majority of the data gaps occurring during low wind nights.

For calculating daily mean fluxes or annual GHG balances, the gaps in flux time series needed to be filled. The gaps were filled separately with three machine learning (ML) algorithms: random forest (RF), extreme gradient boosting (XGB) and k nearest neighbours (kNN). These three algorithms were selected based on their good performance in filling gaps in flux time series in prior studies (e.g., Goodrich et al., 2021; Irvin et al., 2021; Vekuri et al., 2023). Ensemble median of the three gapfilled time series was then used to estimate annual GHG balances and daily fluxes, whereas the spread between the three estimates was used to evaluate the range of plausible annual GHG balance values. This approach minimizes the uncertainty in annual balances that would otherwise result from the selection of a particular algorithm for gapfilling. However, it is possible that the spread may underestimate the total uncertainty of annual fluxes, since it does not take into account, for example, the contribution of random uncertainty associated with EC observations, and it relates only to uncertainty related to the gapfilling process. "xgboost" (version 1.7.1) Python package was used for the XGB method, whereas "scikit-learn" (version 1.1.1) functions RandomForestRegressor and KNeighborsRegressor were utilized in RF and kNN methods, respectively. ML model training and testing of predictive performance was executed as follows: first model hyperparameters were tuned against a random subset of data with scikit-learn function RandomizedSearchCV. After hyperparameter tuning, artificial gaps (covering 15 % of the data) were introduced in random locations in the flux time series and the lengths of these gaps were drawn from a distribution describing the length of actual gaps in the time series (Irvin et al., 2021). Measured data from these gaps were used as independent test data, whereas all the other data were used in model training. Then the trained model predictive performance was evaluated against the test data and this training/testing procedure was executed independently five times. The final models used in gapfilling the flux time series were trained using all the measured data. The following predictors were used in this gapfilling procedure for $CH_4$ fluxes: normalized daily incoming potential solar radiation (RPOT) and its first time derivative, $T_{air}$, its average during the past 3 hours, 1 day and 7 days, incoming shortwave radiation, surface temperature calculated from upwelling longwave radiation, vapor pressure deficit, sine and cosine transformed wind direction and Ts. The list of predictors was the same for $N_2O$ fluxes except also soil water content ($\theta$) was included. For $CO_2$ flux time series gapfilling, otherwise

the same predictors were used as for $CH_4$, but also daily normalized RPOT and its first time derivative (within each day values range between 0 and 1) were included so that the models capture better the $CO_2$ flux daily cycle. For kNN, data were normalized to zero mean and unit variance, whereas for RF and XGB data were not normalized. The predictive performance ($R^2$) of the ensemble models obtained with this procedure were $0.75 \pm 0.04$ (mean $\pm$ standard deviation of the five predictions), $0.66 \pm 0.05$, and $0.92 \pm 0.01$ for $CO_2$, $CH_4$, and $N_2O$ fluxes, respectively, the slopes of linear fits between the predictions and

observations were $0.97 \pm 0.08$, $1.01 \pm 0.02$ and $1.01 \pm 0.04$ and the intercepts were $0.01 \pm 0.07$ µmol m$^{-2}$ s$^{-1}$, $-0.01 \pm 0.04$ nmol m$^{-2}$ s$^{-1}$ and $-0.02 \pm 0.04$ nmol m$^{-2}$ s$^{-1}$, respectively. These results for the predictive performance differ from some of the aforementioned studies, however this is likely due to the nature of variability of these fluxes at our site (low photosynthesis and $CO_2$ flux variability, marked seasonal variability in $N_2O$ fluxes and low $CH_4$ fluxes, see Sect. 3.1).

$CO_2$ fluxes (net ecosystem exchange, NEE; with positive sign denoting net emissions) were decomposed to ecosystem respiration ($R_{eco}$) and gross primary productivity (GPP) following the nighttime decomposition method by Reichstein et al. (2005) with the slight modifications by Wutzler et al. (2018). However, in contrast to Reichstein et al. (2005), here we forced nighttime GPP to zero by subtracting 1.5 day running median of the nighttime GPP from the GPP time series (and added it to $R_{eco}$ time series) and forced any residual nighttime GPP to zero. This way NEE $= R_{eco} -$ GPP is valid at all time steps and

GPP is zero when there is no incoming solar radiation.

### 2.4 EC flux footprint

Turbulent fluxes measured with EC relate to the surface fluxes via

$$F(t) = \iint \varphi(x,y,t) f(x,y,t) \mathrm{dxdy}, \tag{1}$$

where $F = F(t)$ is the flux measured with EC at time t, $f = f(x,y,t)$ is the surface flux at location $(x,y)$ at time $t$ and $\varphi = \varphi(x,y,t)$ is so-called footprint function which describes the source area of EC flux measurements (Vesala et al., 2008) Footprint gives an estimate for the relative contribution of each location on the surface to the measured turbulent flux and with such information it is possible link surface features to measured fluxes. If we assume constant fluxes ($f_j(t)$) from surface-type

$j$ during time step $t$, then Eq. 1 can be simplified as

$$F(t) \approx \sum_j \varphi_j(t) f_j(t),$$

$$\tag{2}$$

where $\varphi_j$ is the overall contribution of surface-type $j$ to the EC flux source area during time step t.

In this study, the source area, i.e., footprint, for the measured gas fluxes was estimated for each 30-min period with the Kljun et al. (2015) model, which is a simple two-dimensional analytical parameterisation of results obtained with backward

Lagrangian stochastic particle dispersion model (Kljun et al., 2002). The model requires information on the flow, namely wind speed, boundary layer height, Obukhov length, standard deviation of lateral velocity fluctuations, friction velocity and wind direction for rotating the footprint to prevailing direction. All these were measured with the EC equipment, except boundary layer height which was retrieved from ERA5 reanalysis product (Hersbach et al., 2023). In addition to these measurements, footprint calculations require information on EC measurement height ($z$), displacement height ($d$) and surface roughness length ($z_0$). The CC surface is a complex mosaic of different surface-types and vegetation heights with small-scale topography. This variability influences the flow field above the surface and this needs to be accounted for in footprint calculations. To resolve this issue, we opted to use varying values for $d$ in the calculations and estimated them from the EC data via logarithmic wind profile equation similarly as in (Helbig et al., 2016) with the exception that only near-neutral periods were used in this analysis and the estimates for $d$ were bin-averaged in wind direction and 30-day bins before using in footprint calculations for reducing the noise stemming from the uncertain calculation procedure. The estimated values for $d$ ranged between 0.8 and 2.0 m ($5^{th} - 95^{th}$ quantiles of the estimates) during the study period (Fig. S2). $z_0$ was implicitly included in the footprint calculations via the ratio between wind speed and friction velocity (Kljun et al., 2015). With this footprint estimation procedure, we accounted for the effect of temporally and spatially varying surface characteristics on the footprints.

**2.5 Drone imaging**

Orthomosaic of the CC area was generated by using drone images captured on June $8^{th}$ 2022 between 12-14 h using DJI Matrice 210 V2 drone equipped with Zenmuse X7 sensor for RGB and Micasense Altum sensor for multispectral images. Flight altitude was 75 m and images were captured with 95% frontlap and 85% sidelap. The weather conditions were cloudy throughout the flight providing even spectral conditions. The images were georeferenced with 10 ground control points measured with a Timble R12 GNSS device and processed into an orthomosaic and Digital Elevation Model (DEM) using Agisoft Metashape 1.7.3 (Agisoft, 2021). The resulting RGB orthomosaic had a ground sample distance (GSD) of 1.16 cm and multispectral orhomosaic of 3.23 cm

**2.6 Surface type classification**

The land surface classification is based on geographical object-based image classification approach similar to De Luca et al. (2019) and was executed using the drone images (see Sect. 2.5). Orthomosaics from the CC area including RGB, Red Edge (RE) and Near Infrared (NIR) channels were merged with the DEM and segmented by spectral signal Euclidian distance using the Large-Scale Mean-Shift (LSMS) segmentation found in Orfeo Toolbox (Grizonnet et al., 2017). Parameters for LSMS were spatialr = 1, range.r = 5, and minsize = 40. The LSMS segmentation resulted in 1.4 million polygons which enabled detailed segmentation of surface cover elements down to ca. 10 by 10 cm in size.

To classify the segments, training data consisting of samples of different surface types were used to train the Random Forest classifier within the Orfeo Toolbox using the means and variances of R, G, B, RE, NIR and DEM channels inside the polygon.

Random Forest uses multiple decision trees trained on bootstrap sets of training data and the classification is based on majority vote of the decision trees (Breiman, 2001). The training data was manually labelled on the segmented polygon in QGIS software using the RGB image and field surveys. The training data covered 0.27% of the CC area and included even numbers of samples for each surface type distributed evenly across the surveyed area to account for small spectral changes due to slight changes in cloud optical depth during the flight. The classes (Table 1, Fig. 1, Fig. S3) were selected by prior field surveys to be a representative set of different surface types that could accurately be distinguished from the drone orthomosaic and were readily identifiable *in situ*. With the trained model, the rest of the segments were classified into different land cover types with mean balanced accuracy of 81.2% and Cohen's kappa coefficient of 0.64. As the number of segments in the different surface-types varies widely, Table 1 shows also the User's Accuracy for class samples. Piles of harvest residue and dead wood are common throughout the area, and in many cases the difference between those classes is difficult to distinguish and the classification can be mixed. A moderate amount of precipitation occurred just before the flight, but this did not affect the classification of exposed peat, despite the presence of small pond in the depressions. The plant-covered ditches, however, can in some cases be classified as the bottom or the field layer.

**Table 1: Surface type classification.** Names of the surface-types, their definition, share of the clearcut area and average footprint area, mean classification confidence level (share of votes for the majority class) of the Random Forest classifier, and the User's accuracy (share of correctly classified segments from the classification) for equal-sized sample of each class.

| Surface type | Definition | Share of clearcut area (%) | Mean share of footprint area (%) | Mean classification confidence (%) | User's accuracy (%) |
|---|---|---|---|---|---|
| Dead wood | Tree trunks and connected branches | 22.8 | 22.6 | 47.8 | 55.6 |
| Harvest residue | Piles of branches left from clearcutting | 7.9 | 3.7 | 53.6 | 82.2 |
| Exposed peat | Peat piles for spruce saplings | 29.0 | 40.6 | 76.7 | 75.6 |
| Litter | Bare dry ground, conifer shoots | 19.9 | 13.5 | 52.2 | 66.7 |
| Bottom layer (mosses) | Mosses, small shrubs | 1.4 | 0.6 | 28.2 | 51.1 |
| Field layer | Small plants | 11.8 | 9.3 | 49.7 | 68.9 |
| Living tree | Larger trees (>ca. 0.5 m) | 4.2 | 5.2 | 66.8 | 82.2 |
| Plant-covered ditch | moss, sedge or other vegetation covering the ditch | 1.1 | 2.4 | 41.2 | 73.3 |
| Ditch (water surface) | Open water surfaces | 1.9 | 2.1 | 70.9 | 95.4 |

## 2.7 Correlation analysis

To understand which environmental parameters should be included in the statistical flux models, we quantified the GHG flux correlation with environmental variables with the bivariate Spearman rank correlation coefficient ($r_s$). $r_s$ was calculated for the 30-min, non-gap filled timeseries (except for GPP) by omitting those 30-min intervals which did not have observations recorded. For $CO_2$ flux we present both the NEE ($F_{CO2}$) and GPP since the NEE consists of two components (GPP and $R_{eco}$). The environmental variables are precipitation (P), PAR, water content in air ($w_{H2O}$), WTD, air and soil temperature ($T_{air}$, $T_{soil}$) and soil water content ($\theta$). $T_{air}$, P, $w_{H2O}$ and PAR are measured at the EC tower and the locations of $T_{soil}$, WTD and $\theta$ measurements are shown in Fig. 1.

## 2.8 Splitting CH₄ and N₂O flux into surface-type and environmental controls

We developed a statistical model that can capture the spatiotemporal variability of the fluxes, $F_{CH4}$ and $F_{N2O}$. We included the surface-type (ST; Table 1) temperature and soil water availability effects to the model. Soil water availability was only included as a general term without ST specific contribution. We opted to use only air temperature as the single independent, ST specific environmental variable in our model since $T_{air}$ can be expected to be uniform across whole clearcut area. The same assumption is more challenging to justify for soil temperature, soil moisture or WTD, which are influenced by soil processes and topography and expected to vary spatially within the study site.

Six alternative models were fitted to the EC flux measurements. The response variable in both models was the natural logarithm of observed fluxes, either CH₄ or N₂O. The first model (Eq. 3) is referred to as baseline model and assumes coherent responses of soil fluxes across the site. The second model (Eq. 4) is an extension of the baseline model and includes a soil moisture ($\theta$) term. The second model is referred to as the baseline $\theta$ model. The third model (Eq.5) is a ST specific model and allows soil-cover specific variation in fluxes similar to Ludwig et al. (2024) and in their temperature responses. Models 4-6 are modifications of the third model. Model 4 (Eq. 6) does not have a ST specific temperature term, model 5 (Eq. 7) includes a soil moisture term and model 6 (Eq. 8) is similar to model 5 but does not include the ST specific temperature term. Models 3-6 (Eqs. 5-8) are named full, full no $\delta$, full $\theta$ and full $\theta$ no $\delta$ models, respectively.

$$\ln(F_i) = \alpha + \beta \frac{T_{air,i} - T_{ref}}{10°C}$$

(3)

$$\ln(F_i) = \alpha + \beta \frac{T_{air,i} - T_{ref}}{10°C} + \zeta \theta_i$$

(4)

$$\ln(F_i) = \alpha + \beta \frac{T_{air,i} - T_{ref}}{10°C} + \sum_{j=1}^{N} \varphi_{i,j} \left( \gamma_j + \delta_j \frac{T_{air,i} - T_{ref}}{10°C} \right)$$

$$(5)$$

$$\ln(F_i) = \alpha + \beta \frac{T_{air,i} - T_{ref}}{10°C} + \sum_{j=1}^{N} \varphi_{i,j} \gamma_j$$

$$(6)$$

$$\ln(F_i) = \alpha + \beta \frac{T_{air,i} - T_{ref}}{10°C} + \zeta \theta_i + \sum_{j=1}^{N} \varphi_{i,j} \left( \gamma_j + \delta_j \frac{T_{air,i} - T_{ref}}{10°C} \right)$$

$$(7)$$

$$\ln(F_i) = \alpha + \beta \frac{T_{air,i} - T_{ref}}{10°C} + \zeta \theta_i + \sum_{j=1}^{N} \varphi_{i,j} \gamma_j$$

$$(8)$$

where $F_i$ is the observed 30-min flux , $\alpha, \beta, \gamma, \delta$ and $\zeta$ are free parameters to be estimated, $\varphi_{i,j}$ is the fraction of surface-type $j$ inside the footprint of observation $i$, $T_{ref} = 10°C$ is the reference temperature and $\theta$ is the mean soil moisture measured at three locations shown in Fig. 1.

For the ST specific models (Eqs. 5-8), we consider models with either 3, 4, 5, 6, or 9 surface-types (ST3, ST4, ST5, ST6 and ST9 respectively) bringing the total number of models that are considered to 22 for both GHG. ST9 considers all the classified surface-types. ST5 is built from ST9 by leaving out the bottom layer class which covers only ca 0.6% on average of the footprint areas and by combining dead wood and harvest residue classes, ditches with water surface (open ditches) and plant-covered ditch classes and living trees and field layer classes. Similarly, ST3 is built from ST5 by further combining all classes except exposed peat class and the class containing both ditch types. ST4 and ST6 are derived from ST3 and ST5, respectively, by allotting the open ditches and plant-covered ditches to their own classes. The different surface-type combinations are summarized in Table 2 (see also Fig. 1 and Fig. S3 for visualization of surface-types in the CC area).

**Table 2: Surface type combinations between ST specific models.** The table shows which surface-types are combined in different ST specific models. Same number in a column indicates that the surface-types are combined. X indicates that the surface-type is removed from the analysis.

| Surface type | ST3 | ST4 | ST5 | ST6 | ST9 |
|---|---|---|---|---|---|

|  |  |  |  |  |  |
|---|---|---|---|---|---|
| Dead wood | 1 | 1 | 1 | 1 | 1 |
| Harvest residue | 1 | 1 | 1 | 1 | 2 |
| Exposed peat | 2 | 2 | 2 | 2 | 3 |
| Litter | 1 | 1 | 3 | 3 | 4 |
| Bottom layer (mosses) | X | X | X | X | 5 |
| Field layer | 1 | 1 | 4 | 4 | 6 |
| Living tree | 1 | 1 | 4 | 4 | 7 |
| Plant-covered ditch | 3 | 3 | 5 | 5 | 8 |
| Ditch (water surface) | 3 | 4 | 5 | 6 | 9 |

The free parameters of the models $\alpha, \beta, \zeta, \gamma_j, \delta_j$ and $\sigma_\epsilon$ (the standard deviation of the likelihood function) were estimated using Bayesian inference and Markov chain Monte Carlo (MCMC) methods using the "pyMC" package (Abril-Pla et al., 2023). The prior distribution of $\alpha$ and $\zeta$ was set to a normal distribution whose mean was zero and standard deviation was two. The prior distribution of $\gamma$ follows a hierarchial design: the prior for each surface-type is normally distributed with mean $\mu_\gamma$ and standard deviation $\sigma_\gamma$ and the prior distribution for mean $\mu_\gamma$ was the standard normal distribution $\mu_\gamma \sim \mathcal{N}(0,1)$. We used exponential

distributions as priors for $\beta$ and $\delta$ with rate parameters $\lambda_\beta$ and $\lambda_\delta$. We used a normally distributed likelihood function with standard deviation $\sigma_\epsilon$. We set the prior distribution for $\sigma_\epsilon$ to be exponential distribution with rate parameter $\lambda_\epsilon$. Finally, the rate parameters $\lambda_i$ of the exponential distributions for $\beta, \delta, \sigma_\epsilon$ and $\sigma_\gamma$ were set such that the full width at half maximum (FWHM) of prior predictive distributions (Fig. S4-S5) is at least 2 times wider than the FWHM of the observed flux distributions. For simplicity, same values were used for both GHGs $\sigma_\gamma = 2.0, \lambda_l = 1.0; l \in \{\beta, \delta, \epsilon\}$. The parameters were

estimated using the *pymc.sampling.mcmc.sample* function of the pyMC package with 4 chains, 2000 samples per chain and a tuning period of 2000 steps, i.e., total 8000 individual parameters sets were drawn for further analysis. All the other sampler settings were left as default. The full, non gap-filled, EC flux data sets were used in the parameter estimation i.e., the artificial gaps introduced to the flux data sets for developing the gap-filling model were not present in this parameter estimation.

We evaluate the model performance based on the expected log posterior density of leave one out cross validation (ELPD-LOO). ELPD-LOO was calculated using the compare function of the "ArviZ" Python package which uses the Pareto smoothed importance sampling to re-fit the model parameters (Vehtari et al., 2017). The compare function ranks the models based on the expected log posterior density of the left out samples. While a single ELPD-LOO value is not easy to interpret in terms of model performance, models are straightforward to compare against each other as higher value of ELPD-LOO marks better

performance.

We defined Eqs. (3)-(8) on natural logarithm base since ln-transformations of the measured flux values were normally distributed based on quantile-quantile plotting. When transforming the measured 30-min fluxes to natural logarithm base, we omitted those $CH_4$ fluxes that were below $-10$ nmol m$^{-2}$s$^{-1}$ and those $N_2O$ values that were below 0 nmol m$^{-2}$s$^{-1}$. We chose these limits since during low flux period $CH_4$ fluxes varied randomly around zero, whereas $N_2O$ fluxes were clearly positive throughout the measurement period with only occasional negative flux observations. The $CH_4$ flux values were then shifted by 10 nmol m$^{-2}$s$^{-1}$ before the natural logarithm was applied. This shift was accounted for also when the model results were transformed back into units of nmol m$^{-2}$s$^{-1}$. Additionally, we accounted for natural logarithm transformation bias when transforming the modelled fluxes to nmol m$^{-2}$s$^{-1}$. In total the back transformation is

$$F_i = \exp\left(F_{i,ln} + \frac{\sigma_\epsilon^2}{2}\right) - S,$$

(9)

where $F_{i,\ln}$ is the modelled flux in natural logarithm base, $\sigma_\epsilon$ is the standard deviation of the likelihood function and $S$ is the shift ($S = 0$ nmol m$^{-2}$s$^{-1}$ for $N_2O$ and $S = 10$ nmol m$^{-2}$s$^{-1}$ for $CH_4$)

To further understand the GHG emissions from different surface-types, we calculated the surface-type specific fluxes by setting the contribution of each surface-type to unity ($\varphi_{i,j} = 1$ in Eq. 4) in turn, while zeroing others. We calculated 8000 different flux values with the parameters estimated in the MCMC sampling.

## 3 Results

### 3.1 Ecosystem scale greenhouse gas fluxes

The CC area in the Ränskälänkorpi study site was a strong net source of GHGs during the first full year (second growing season) after the clearcutting (Fig. 2 and Fig. S6). The eddy covariance measurements showed that the $CO_2$ was the dominant GHG flux in terms of emissions (expressed as $CO_2$-equivalents, $GWP100_{CH4} = 27$ , $GWP100_{N2O} = 273$ ;Forster et al., 2023). Specifically, the annual NEE was controlled by $R_{eco}$ (38200 kg $CO_2$-eq ha$^{-1}$ yr$^{-1}$ ; 1040 g C m$^{-2}$ yr$^{-1}$ $34900 - 40300$ kg $CO_2$-eq ha$^{-1}$yr$^{-1}$ depending on the EC gap-filling method) rather than by GPP (14900 kg $CO_2$-eq ha$^{-1}$ yr$^{-1}$ ; 410 g C m$^{-2}$ yr$^{-1}$ $12500 - 16200$ kg $CO_2$-eq ha$^{-1}$yr$^{-1}$ ) . Consequently, the CC area showed high cumulative net $CO_2$ emissions during 2022 ( 23300 kg $CO_2$-eq ha$^{-1}$ yr$^{-1}$ ; 640 g C m$^{-2}$ yr$^{-1}$ ; $22400 - 24100$ kg $CO_2$-eq ha$^{-1}$ yr$^{-1}$ ), followed by relatively low $N_2O$ emissions ( 5000 kg $CO_2$-eq ha$^{-1}$ yr$^{-1}$; $4900 - 5100$ kg $CO_2$-eq ha$^{-1}$yr$^{-1}$) and only minor $CH_4$ emissions (100 kg $CO_2$-eq ha$^{-1}$ yr$^{-1}$; 0.3 g C m$^{-2}$ yr$^{-1}$). It should be noted that the contribution of the snow-free period emissions to annual emissions were 82%, 80%, and 98% for $CO_2$, $N_2O$ and $CH_4$, respectively.

The seasonal cycle of NEE was characterized by small emissions ($R_{eco}$) during winter. $R_{eco}$ increased rapidly after snowmelt, while GPP remained low until late May. The NEE was rather stable from late May to late Aug, while both component fluxes showed dual peak in late June and August. In autumn, GPP decreased along the reduced solar radiation but respiration remained at nearly constant level from September to November, causing the seasonal asymmetry seen in NEE (Fig. 2). On daily scale, the ecosystem was a net source of $CO_2$ to the atmosphere throughout the measurement period. $CH_4$ flux started to increase in the mid-June slightly over one month after snow melt and the daily $CH_4$ emissions fluctuated between $1 - 6$ nmol m$^{-2}$s$^{-1}$ until the end of August after which the flux was small ( $-1 - 1.6$ nmol m$^{-2}$s$^{-1}$ ). $N_2O$ flux increased from 0.5 to 1.5 nmol m$^{-2}$s$^{-1}$ from mid-April to mid-May and after a short decrease, it gradually increased to 2 nmol m$^{-2}$s$^{-1}$ by mid-July. Between mid-July and mid-August $N_2O$ flux experienced a strong peak with highest values of 6 nmol m$^{-2}$s$^{-1}$. $N_2O$ flux stayed then around 2 nmol m$^{-2}$s$^{-1}$ until the snow covered the clearcut area after which the flux decreased below 1 nmol m$^{-2}$s$^{-1}$.

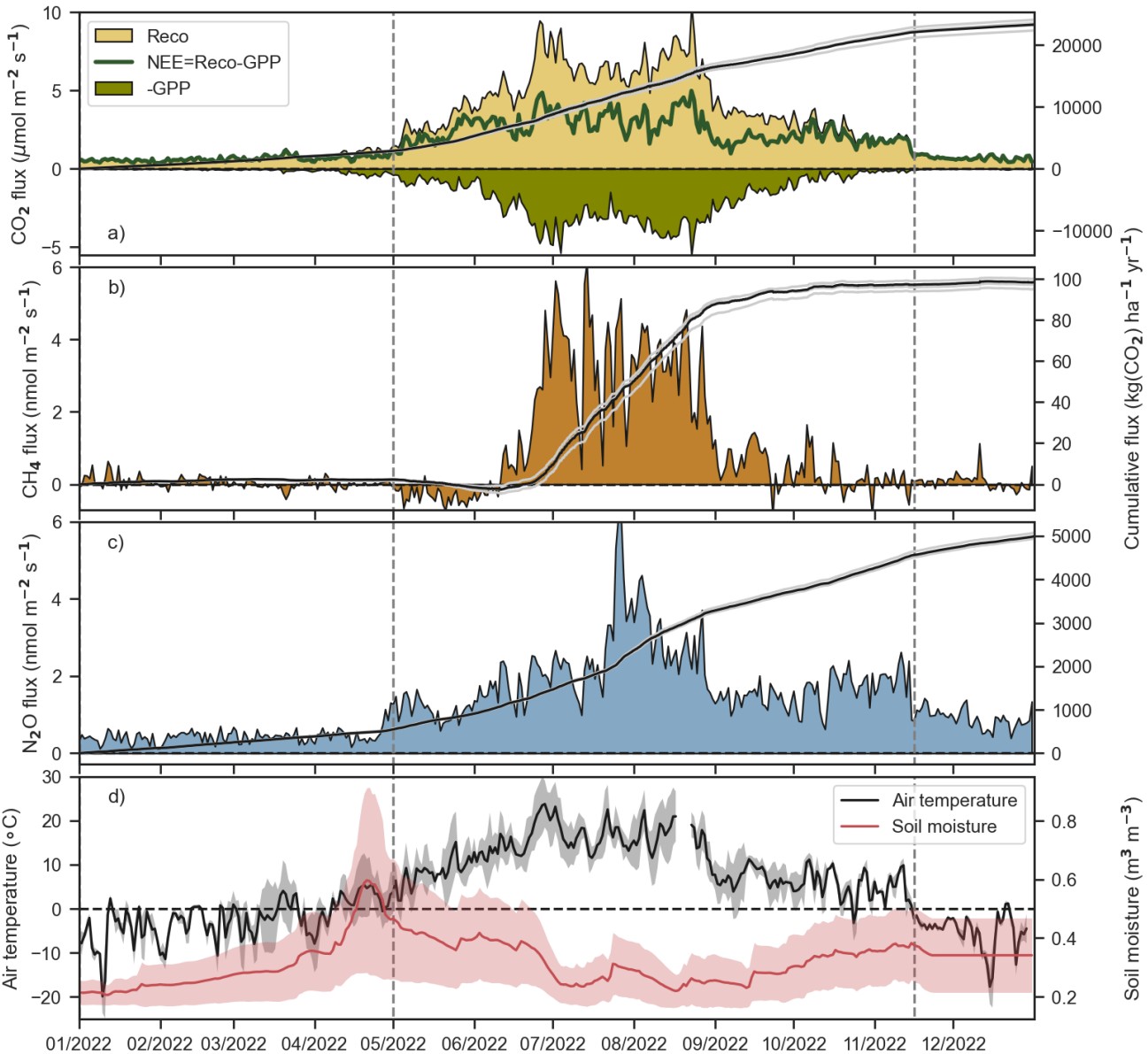

**Figure 2. Time series of daily mean and cumulative sums of $CO_2$ (a), $CH_4$ (b) and $N_2O$ (c) fluxes, daily air temperature and soil moisture (d) during the year 2022.** $CO_2$ flux is partitioned into components of gross primary production (GPP) and ecosystem respiration ($R_{eco}$) with methods described in Sect. 2.3. Vertical dash lines indicate the snow melt dates in spring and first snow in late autumn. Flux time series were gapfilled with three different ML algorithms (Sect. 2.3) and cumulative sums calculated from these time series are shown with
440 grey lines. Black line shows the ensemble average of these three estimates. Shaded area in panel d shows daily temperature and soil moisture variability (standard deviation) around the mean.

## 3.2 Flux correlation with environmental parameters

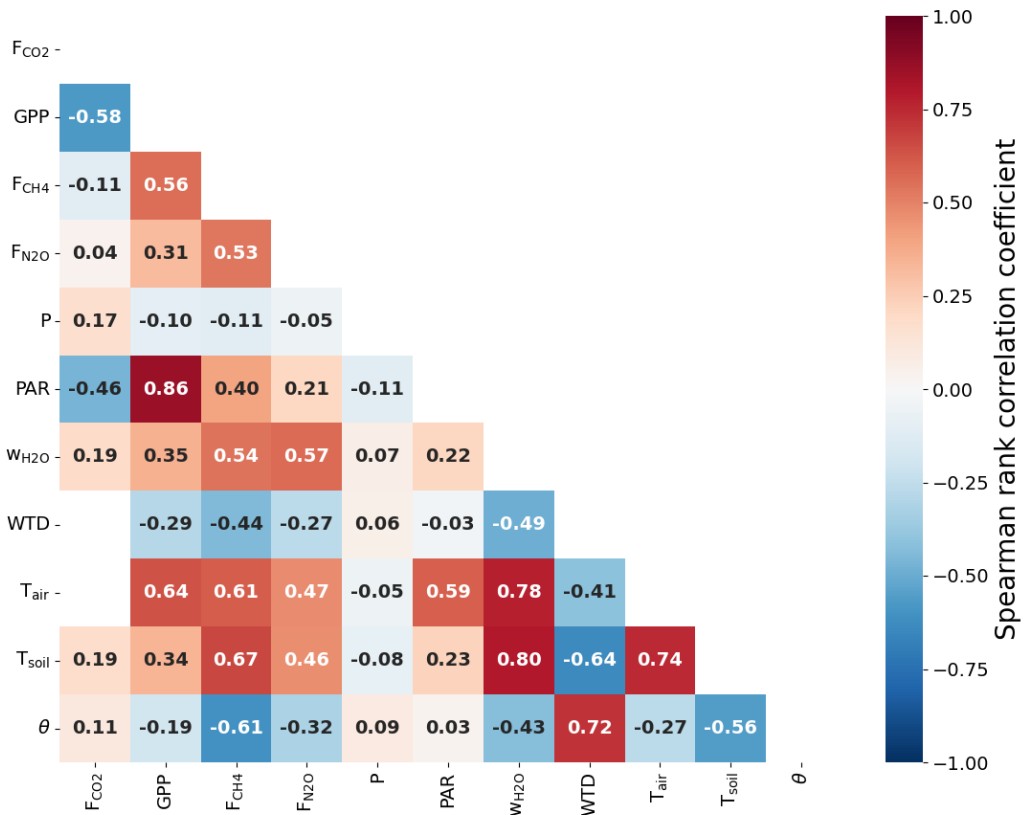

**Figure 3. Correlation heatmap reporting Spearman's rank correlation coefficients for GHG fluxes and selected environmental parameters.** The abbreviations are: $F_{CO2}$ = CO$_2$ flux, $GPP$ = gross primary production, $F_{N2O}$ = N$_2$O flux, $F_{CH4}$ = CH$_4$ flux, $P$ = precipitation, $PAR$ = photosynthetically active radiation, $w_{H2O}$ = water mixing ratio in air, $WTD$ = water table depth, $T_{air}$ = air temperature, $T_{soil}$ = soil temperature at 5 cm depth (averaged value obtained from 3 different sensors located over the CC area; see white circles in Fig. 1), $\theta$ = soil water content at 5 cm depth (average value similar to $T_{soil}$). For further details on the measurement locations of other parameters, please refer to Fig. 1 and Sect. 2.2. Only correlations whose p-value is lower than 0.05 are shown.

Figure 3 shows correlation coefficients between the 30-minute GHG fluxes and environmental variables. The NEE correlated well ($|r_s| > 0.25$) only with PAR while the GPP correlates with PAR, $w_{H2O}$ and both $T_{air}$ and $T_{soil}$. $F_{CH4}$ correlated with all environmental variables besides P. The $F_{CH4}$ correlated positively with temperature, $w_{H2O}$ and $PAR$, and negatively with $WTD$ (i.e., higher $F_{CH4}$ are observed when WTD is close to the surface) and $\theta$. $F_{N2O}$ correlation with environmental factors was similar to $F_{CH4}$ except that it correlated weaker with PAR, WTD, $\theta$, $T_{air}$ and $T_{soil}$. Most of the environmental variables are correlated with each other due to their similar diel and annual cycles.

### 3.3 Models for CH₄ and N₂O fluxes to estimate surface-type specific fluxes

The performance of each model are shown in Table S1. We selected the best models (based on) for further analysis: Full $\theta$ ST9 for both $F_{N2O}$ and $F_{CH4}$

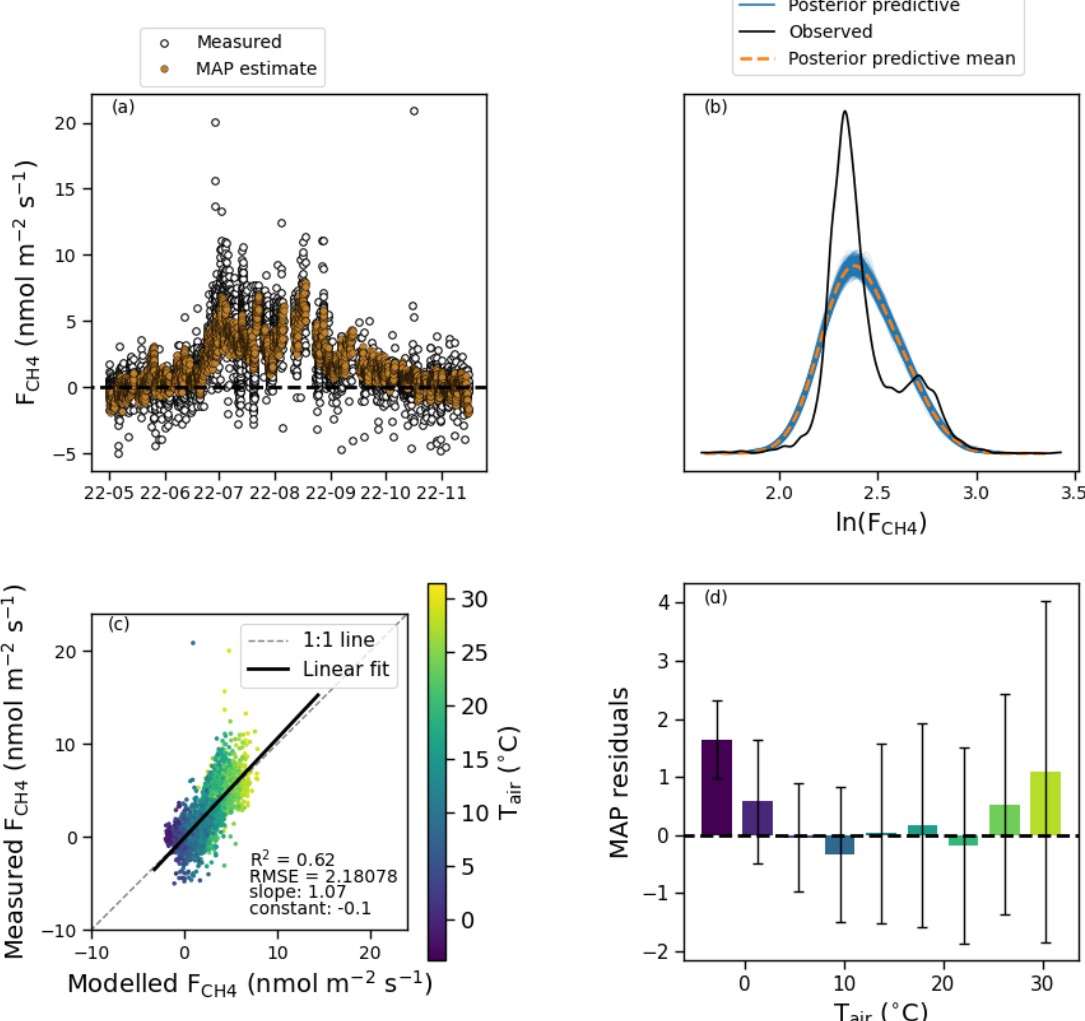

**Figure 4. Time series of measured and modelled CH₄ flux (a), distribution of measured CH₄ flux and the posterior predictive distributions (b), scatter plot of modelled versus measured CH₄ flux (c) and the model residuals as function of air temperature (d).** The model estimates are calculated with the maximum a posteriori (MAP) estimate of the parameters.

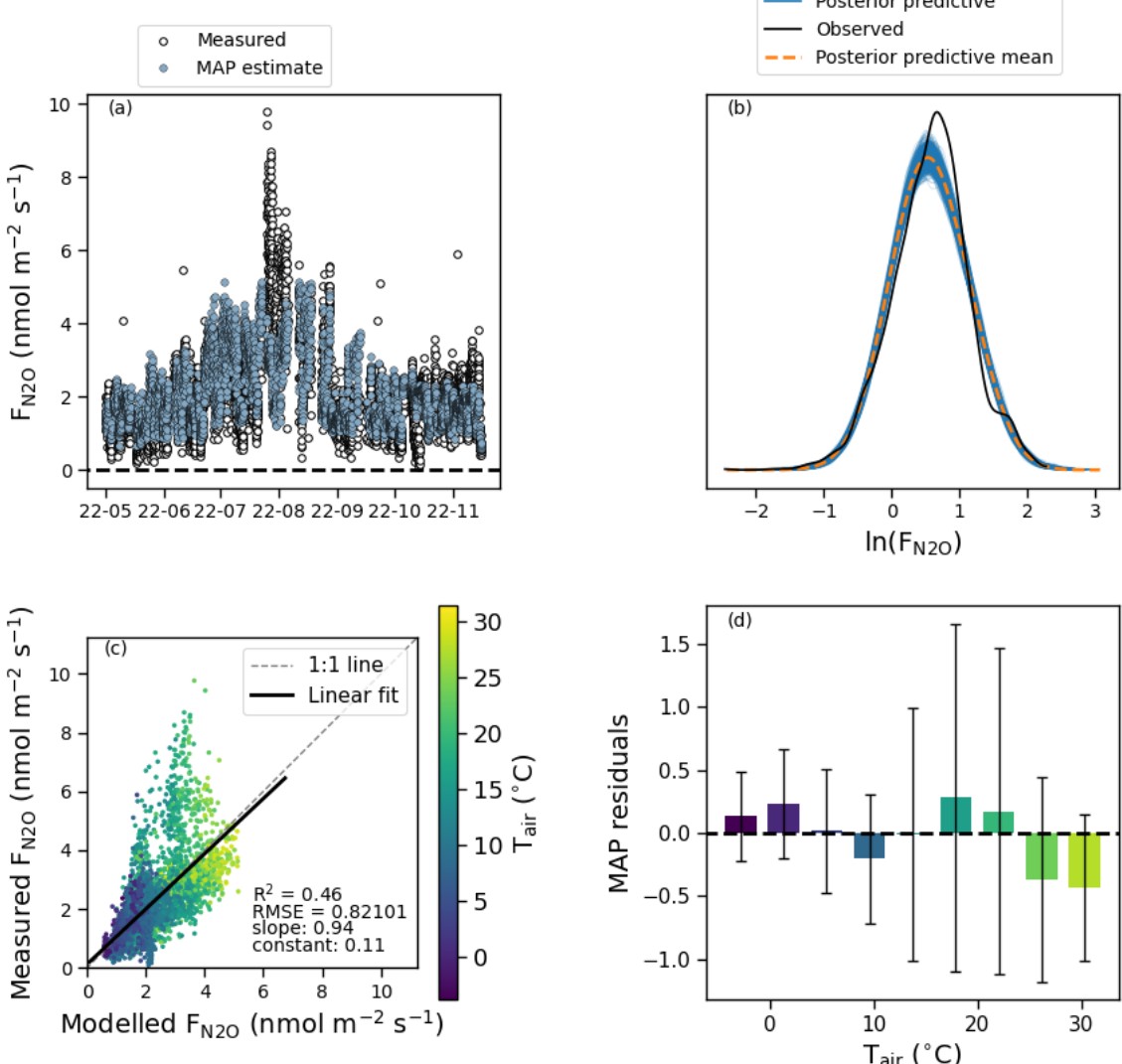

**Figure 5. Time series of measured and modelled N₂O flux (a), distribution of measured N₂O flux and the posterior predictive distributions (b), scatter plot of modelled versus measured N₂O flux (c) and the model residuals as function of air temperature (d).** The model estimates are calculated with the maximum a posteriori (MAP) estimate of the parameters.

For CH₄ the posterior predictive distribution (Fig. 4b) of the best model showed that the model both over and underestimated the measurements, which were distributed very narrowly with two peaks at $\ln(F_{CH4}) = 2.35$ ( $0.5$ nmol m$^{-2}$s$^{-1}$ )and $\ln(F_{CH4}) = 2.75$ ($5.6$ nmol m$^{-2}$s$^{-1}$). The best model for CH₄ could captures 62% of the variation in the measurements. The model parameters (Fig. 6 a-b) indicate the flux has weak temperature dependency except for ditches, high base source strength ($\alpha$) and low surface-type specific base strength modifier ($\gamma$) except for plant-covered ditches. This suggests that there are no major differences in source strengths between the surface-types, expect for the plant covered ditches from which the emissions are clearly higher than from other parts of the CC area.

The posterior predictive distribution for the best $F_{N2O}$ model shows a better fit to the observations (Fig. 5b) but fails to capture the peak N₂O emissions observed at the end of July (Fig. 5a). The $R^2$ value between modelled and measured flux is 0.46, lower than for the best $F_{CH4}$ model. The best $F_{N2O}$ model indicates higher variation between fluxes from different surface-types than the model for $F_{CH4}$ (Fig. 6c). Similarly, the temperature dependency defining parameter $\delta$ varies more between different surface-types than it did for $F_{CH4}$ (Fig. 6d). The model residuals (calculated with the MAP estimate) for both GHGs do not show a clear dependency of the air temperature (Fig. 4d and Fig. 5d) indicating no clear over- or underfitting with respect to certain temperature range.

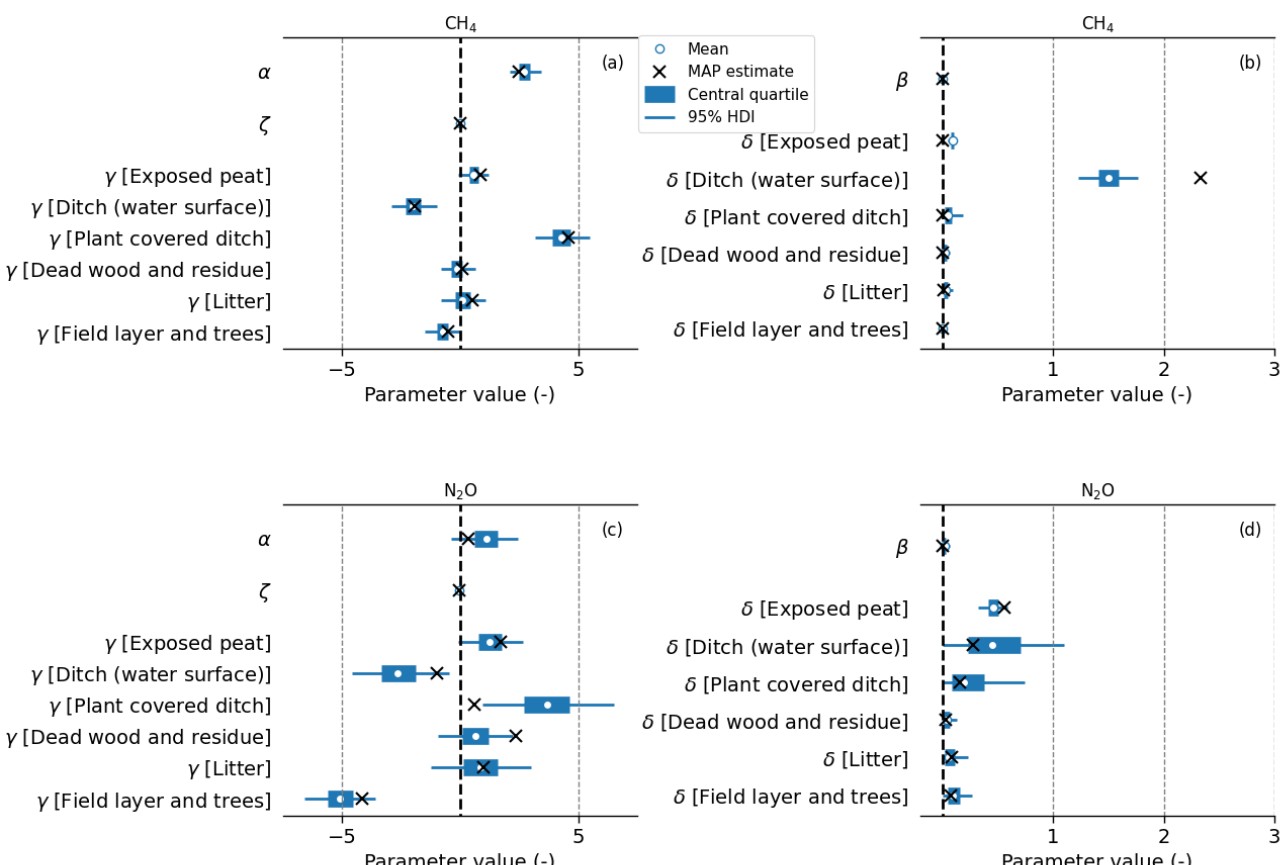

**Figure 6. 95% highest posterior density interval for parameters of the best models for CH₄ (a-b) and N₂O (c-d).** The bold line indicates the 25th and 75th percentiles of the distributions, white circles are the distribution means and black crosses show the maximum a posteriori (MAP) estimate.

Fig. 7 shows the distribution of the modelled surface-type specific fluxes, predicted by setting the corresponding surface-type contribution to unity ($\varphi_{i,j} = 1$ for each $j$ in Eq. 7) and calculating the 95% highest density interval of the resulting model using the measured $T_{air}$. The results are extrapolations of the underlying model to visualize the model parameters in Fig. 6.

The highest CH₄ emissions originate from plant-covered ditches (Fig. 6a, Fig. 7a), while emissions from the exposed peat and litter are over an order of magnitude smaller. The other surface-types show small uptake and emissions of CH₄. Living trees

and litter show highest N₂O emissions (Fig. 6c-d, Fig. 7 c-d). The second highest N₂O emission come from dead wood and plant covered ditches.

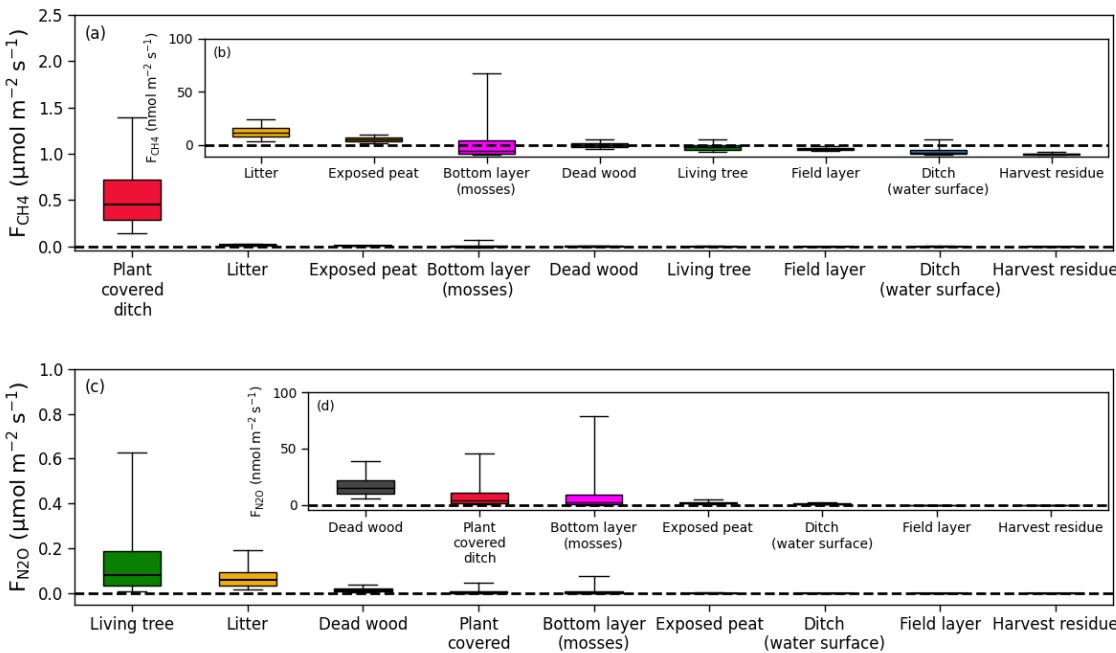

**Figure 7. Surface type specific flux of CH₄ (a-b) and N₂O (c-d) fluxes.** The surface-type specific fluxes are calculated by setting the corresponding surface-type contribution to unity ($\varphi_{i,j} = 1$ for each $j$ in Eq. 7) and calculating the 95% highest density interval of the resulting model with the measured $T_{air}$. The boxplot whiskers represent 5th and 95th percentile of the flux value distribution, the edges of the post represent 25th and 75th percentile and the black horizontal line shows the median of the distribution. The boxplots are ordered by the 75th percentile. Note the different scales of the y-axis for each panel and that the fluxes are per square meter of the respective surface type. To estimate the contribution of each surface type to the total flux values in this figure need to be multiplied by the fraction of the surface type in the clearcut area or in the EC footprint presented in Table 1.

Fig. S7 and Fig. S8 show how the modelled flux changes when surface-types are added one by one to the model and how the model results agree with chosen measurements. From the analysis it is evident that the most important surface-types for the footprint-average CH₄ emissions are the plant-covered ditches (areal coverage 1.1%, Table 1) and exposed peat (29%). For N₂O emissions the most important surface-types are exposed peat, litter (19.9%), dead wood (22.8%) and field layer (11.8%).

Fig. S9-S12 show the estimated parameters for the full $\theta$ models for the other number of STs. Interestingly, for $CH_4$ when the two types of ditches are lumped into one ST, their $\gamma$ estimate is close to zero (Fig. S9 and S11) whereas when the ditches are

considered as separate STs the estimated $\gamma$ for the plant-covered ditch is the highest and the $\gamma$ for the ditches with water surface is the lowest which is the same behaviour what we see in Fig. 6 for the best model.

The parameter estimates between different number of STs for $N_2O$ models differ more than for $CH_4$ models. For example for ST6 (Fig. S12) the highest $\gamma$ MAP estimate is for dead wood and residue whereas the $\gamma$ for the field layer and trees is the

smallest. The $\gamma$ estimates for ST5 (Fig. S11) seem to also emphasize the role of litter and dead wood and residue as high $N_2O$ emitting surface types. It should be noted that for all other number of STs the living trees are always lumped together with some other surface type or types. It might be for this reason that the full $\theta$ no $\delta$ ST9 model outperforms the full $\theta$ ST6 model clearly in $N_2O$ but not for $CH_4$ (Table S1).

Finally, we calculated the total emissions for $CH_4$ and $N_2O$ for the snow free period using the best full $\theta$ models and compared them to EC measurements (Table 3). The model estimates were calculated using the areal fraction of each surface type in the whole clearcut (Table 1), instead of areal fraction of the surfaces in the EC footprints. This way the reported modelled flux estimates are representative for the whole clearcut area. The predicted cumulative $CH_4$ emission is 60% smaller than that based on EC whereas for $N_2O$, the emissions from EC are ca. 1.25 times higher than the median model prediction. This might be

either due to model inaccuracies (see Figs. 4 and 5) or due to the EC observing a biased sample of the ecosystem-atmosphere exchange (i.e. certain surface types have higher/lower areal coverage in EC footprints than they have in the whole clearcut, see Table 1). However, the EC derived emission estimate is inside the 95% highest density interval for both GHGs.

**Table 3. Comparison of methane and nitrous oxide emissions obtained from the EC measurements and predicted for the whole**
**clearcut area by the models that best described the temporal variability of fluxes.** Note that the snow-free period is from 1[st] May to 16[th] November. For the modelling approach, the first value represents the median model prediction, while the values in brackets present the 95% highest density interval of distribution of the snow-free period emissions calculated with the parameters estimated in the MCMC run. For the EC results, the values show the total emissions calculated from time series gapfilled with ML ensemble, while the values in brackets show the range of values calculated from time series gapfilled with different ML algorithms (see Sect. 2.3). Note that the share of each
surface type in the whole clearcut were used to calculate the modelled flux estimates and hence they relate to the whole clearcut area, not to the EC footprint area.

| Greenhouse gas | Modelling approach snow free period (kg $CO_2$-eq. ha$^{-1}$) | Eddy covariance snow free period (kg $CO_2$-eq. ha$^{-1}$) | Eddy covariance full year (kg $CO_2$-eq. ha$^{-1}$) |
|---|---|---|---|
| $CO_2$ | - | 19200 (18400 - 20000) | 23300 (22400 - 24100) |
| $CH_4$ | 40 (-140 - 240) | 100 (100 - 100) | 100 (100 - 100) |
| $N_2O$ | 3200 (1200 - 6400) | 4000 (3900 - 4100) | 5000 (4900 - 5100) |

## 4 Discussion

### 4.1 Impact of surface types on CH$_4$ and N$_2$O fluxes

We built a statistical model to separate observed CH$_4$ and N$_2$O fluxes into their surface-type and environmental controls using the flux timeseries and surface type composition for each measurement period inferred from drone-based surface characterization and analytical footprint model. The aim of the analysis was to assess whether the fluxes vary across different surface types, and to detect the key surface types contributing to the net emissions. The models suggest that plant-covered ditches and exposed peat are the most important surface-types for CH$_4$ emissions (Fig. 6, Fig. 7, Fig. S7), while other surface-types contributed much less or acted as CH$_4$ sinks. This is consistent with chamber measurement at this study site, which showed that the soils acted as CH$_4$ sinks, as the aerobic soil layer above the water table is able to consume CH$_4$ from both the deep soil and the atmosphere. The high CH$_4$ emissions observed in ditches can be attributed to two main factors: the high production of CH$_4$ in anaerobic ditch sediments and the transport of CH$_4$ from surrounding soils by drainage water. Rissanen et al. (2023) found that ditches with open water exhibited higher emissions than those covered by plants. In particular, ditches covered by mosses showed very low emissions, as CH$_4$ can be oxidized in the moss layer. Minkkinen and Laine (2006) reported that the CH$_4$ emissions from ditches varied considerably depending on the water movement and vegetation cover. They found that ditches with moving water showed higher emissions, likely due to the transportation of CH$_4$ from the surrounding areas. However, we found that the plant-covered ditch surface was the highest emissions source, which probably because the main ditch in close proximity to the EC tower was classified as plant-covered due to the vascular plants growing on the ditch bank (Fig. S1). It contributed the majority of the CH$_4$ emissions according to our analysis. It should be noted that our classification did not distinguish between moss- and vascular plant-covered ditches. In contrast to mosses which can act as a filter for CH$_4$ due oxidation (Kolton et al., 2022; Larmola et al., 2010), some vascular plants, such as *Eriophorum*, can enhance transport of CH$_4$ to the atmosphere (Minkkinen and Laine, 2006). The second largest contributor to CH$_4$ emissions in our model was exposed peat. Pearson et al. (2012) observed that the contrasting effects of mounding with exposed peat on CH$_4$ emissions from soil depends on the drainage condition. Soil CH$_4$ emissions depend on production, consumption, and transportation processes. Production is largely controlled by the WTD, which determines the anoxic layer, whereas the surface types in clearcut with distinct topography and ground vegetation can affect the consumption and transportation.

Measured N$_2$O fluxes showed strong temporal variation over the studied year (Fig. 2). The short periods of high N$_2$O emissions observed during the snow-free period, which contribute considerably to the annual budget, have been previously documented in peatland sites through continuous measurements based on EC and automatic chambers (Pihlatie et al., 2010). Our model was, however, unable to predict the high N$_2$O emission period observed during late July and early August. The high emissions are likely driven by the activity of specific archaea and prevailing conditions, including temperature, moisture, C/N ratio,

nitrate content, pH, and peat decomposition phase (Bahram et al., 2022). Our modelling approach, unfortunately, lacked some of these details. Furthermore, our study demonstrated that $N_2O$ emissions during the snow-covered period constituted a significant (20%) contribution to the annual budget. The winter under study was meteorologically typical, which may explain why the importance of these emissions is consistent with that reported in previous works (Rautakoski et al., 2024; Kim and Tanaka, 2002). However, the frequency of warm winters with lower snow cover and more freeze-thaw events is predicted to increase in northern latitudes (Ruosteenoja and Räisänen, 2021). As a result, snow-covered $N_2O$ emissions are expected to increase in boreal forests. In terms of spatial variability, our model showed that the majority of $N_2O$ emissions were attributable to surfaces with living trees, exposed peat, dead wood and litter (Fig. 6, Fig. 7, Fig. S8). A previous study, which employed chamber measurements, corroborates our modelling findings (Mäkiranta et al., 2012). It was observed that soils in peatland forests covered by logging residues exhibited high $N_2O$ emissions after harvesting, which was attributed to the decay of the logging residuals. Pearson et al. (2012) also found high $N_2O$ emissions from the mounds (surfaces with exposed peat) following site preparation in a nutrient-poor clearcut peatland forest. $N_2O$ emissions were found to be highly dependent on the availability of N in the soil (Ojanen et al., 2010). Therefore, it can be concluded that the observed variation in $N_2O$ emissions from different surface-types may also be related to the spatial variability of nutrient conditions within the studied clearcut area.

Studies of soil microclimate and gas fluxes after clearcutting and site preparation are scarce on drained peatland forests, but the few done using manual chamber method (Pumpanen et al., 2004; Mjöfors et al., 2015; Strömgren et al., 2016, 2017) have showed that the spatial variability is typically very high. Pearson et al. (2012) applied the manual chamber method to assess the impact of varying microtopography following site preparation in a nutrient-poor clearcut peatland forest for $CO_2$, $CH_4$ and $N_2O$. Gas fluxes from ditches can be measured by manual chamber floating on ditch water (e.g., Minkkinen and Laine, 2006; Rissanen et al., 2023). However, ditch banks, where the ditch materials are exposed and may act as $CH_4$ hotspots, have rarely been measured due to the difficulty of installing chambers on uneven surfaces. Our surface-type model could facilitate the understanding of the contribution of surfaces to $CH_4$ and $N_2O$ emissions, particularly those that are not or have been challenging to quantify previously. Furthermore, we identified the surface types that are likely to have high $CH_4$ and $N_2O$ emissions after clearcutting. These surface types should be targeted in future chamber studies to accurately quantify the surface-specific emission fluxes (or emission factors).

### 4.2 Methodological issues and outlook

The models for $F_{CH4}$ and $F_{N2O}$ were found to explain slightly less than 62% and 46%, respectively, of the observed temporal variation. Moreover, the model that best explained the variability in $F_{CH4}$ produced lower cumulative flux over the snow free season for the whole clearcut than what was measured by the EC from its footprint (Table 3). This disagreement might be either due to model inaccuracy or the slightly unrepresentative location of the EC tower (Table 1; see also Chu et al., 2021). However, the estimates were still withing the 95% HDI. The underlying assumptions in our model approach are i) surface type variability drives the variability of soil processes underlying the fluxes, an assumption that can be tested using e.g. chamber

studies, ii) the relative contribution of surface types for each 30-min EC flux can be determined by footprint analysis, and iii) the surface types can be reliably characterized from aerial RGB and multispectral images.

For both $CH_4$ and $N_2O$, we found a clear improvement in model predictions when the effect of surface-types was introduced in the models (Table S1). For $CH_4$ the deviation in model performance between different surface-type specific models was minor, suggesting that the $CH_4$ emissions may be less dependent on the surface-type than $N_2O$ emissions. The Bayesian inference method was selected for its capacity to incorporate prior knowledge into the model. With Bayesian framework, we were able to define the surface-type specific flux strength modifiers (parameters $\gamma_j$) in a hierarchical manner. This resulted in each surface-type having a distinct base production distribution, while the mean of each distribution was derived from a common underlying distribution. Furthermore, there are other types of prior knowledge that could be incorporated to the model to improve the surface-type specific flux estimates. For instance, chamber measurements of surface-type specific flux could be employed to inform the model development, particularly as they could be used as a prior information to constrain the model (Ludwig et al., 2024). Results also revealed that there is a strong negative correlation between $CH_4$ and $N_2O$ flux with soil water availability (Fig. 3), a finding that is consistent with previous observations in drained peatland forests (Ojanen et al., 2010). Inclusion of soil moisture dependent term in the model improved the model's predictive performance. However, only a general soil moisture term was included. As moisture and microtopography vary across the clearcut, distributed measurements of water table (or soil moisture) would be necessary for further refinement of the model. We tested replacing the soil moisture with WTD but the models with WTD performed worse than the full $\theta$ ST9 models for both compounds. Given that we only considered WTD measurement at one location, we cannot be sure that the soil moisture would outperform WTD as an independent variable also with more distributed measurements. Furthermore, the spatial variability of $N_2O$ emissions might be further explained by variables describing nutrient availability (e.g., C:N ratio).

A few previous studies have used surface-type information and EC measurements to elucidate surface-type specific fluxes. In the tundra ecosystem, Tuovinen et al. (2019) and Ludwig et al. (2024) developed a model for $CH_4$ flux by decomposing the total flux into sum of fluxes from different surface-types. In both studies, the models captured more distinct surface type fluxes than our model for $CH_4$. Similarly, in peatlands, Franz et al. (2016) and Forbrich et al. (2011) were able to achieve a better agreement and more distinct surface specific emissions with $CH_4$ modelled from surface-type specific fluxes and EC measurements than our $CH_4$ model. Also Mazzola et al., (2021) found, based on chamber measurements, that there was a clear difference between surface type specific $CH_4$ emissions on a restored bog site in northern Scotland. One possible explanation for this discrepancy is that the surface-types in our model are rather homogeneously distributed in our drained peatland clearcut compared to the other sites, which makes the attribution of fluxes to different surface-types more challenging, as their relative contribution within the flux footprint does not strongly depend on wind direction. Another possibility is that for some of the surface types their proportion inside the footprint is always so low that their contribution to the model estimate is diluted by the surrounding landscape (e.g., ditches with water surface). As a result the model might not correctly capture their contribution

to the flux. Regarding the N$_2$O emissions, we are not aware of any previous studies that have attempted to model surface-type specific fluxes based on EC-data. However, given that N$_2$O was the second largest GHG source from the clearcut area, it is evident that such studies are required in order to improve GHG budget estimation in the future.

For the best models we also tested replacing T$_{air}$ with mean soil temperature measured at the tree locations shown in Fig. 1. For N$_2$O this produced slightly better fit in terms of ELPD-LOO (difference of 74 units). This suggests that especially for understanding N$_2$O emissions, measuring the surface type specific soil temperature would be beneficial.

Footprint calculations were sensitive to the input parameters used in the calculations, hence altering the estimation of surface-type specific CH$_4$ and N$_2$O fluxes. For instance, the displacement height ($d$) was empirically estimated from data (see Sect. 2.3) and changing the estimation procedure altered the footprint results. This was because changes in $d$ directly affects the effective measurement height ($z - d$), which is one of the main factors for the footprint size (e.g., Rannik et al., 2012). These uncertainties essentially stem from the fact that the clearcut surface is heterogeneous, with varying plant height and small-scale topography. The spatial heterogeneity varies with wind direction, and this altered the flow field observed with the EC equipment. Therefore, the estimation of descriptive values for all the parameters needed by the footprint model, e.g., $d$, is uncertain.

Furthermore, it is important to note that simple footprint models, such as the Kljun model used in this study, are only strictly valid above the roughness sublayer, where individual surface roughness elements (e.g., trees, branch piles, etc) do not anymore locally alter the flow. Even above roughness sublayer they rely on simplified theories on the flow field, such as the Monin-Obukhov theory, which are unable to handle e.g., non-stationarities. Nevertheless, such models are utilised also in complex roughness sublayer flows (Chu et al., 2021) to link the observed turbulent fluxes to surface features (Stagakis et al., 2019). It is likely that our EC tower was frequently within the roughness sublayer. Although simple footprint models have been shown to produce reasonable estimates for flux source areas in ideal measurement locations (Arriga et al., 2017; Heidbach et al., 2017; Nicolini et al., 2017; Dumortier et al., 2019; Rey-Sanchez et al., 2022), it is unclear how the estimates are affected by the roughness sublayer flow. The presence of surface roughness elements increases turbulent mixing, which may result in shorter footprints than would be expected for flows above smoother surfaces. Nevertheless, the empirically estimated values for $d$ and $z_0$ may already partly account for this. The methodology used here to derive surface-type specific fluxes did not consider the aforementioned uncertainties. Furthermore, we assumed in the Bayesian framework that the footprints were observed perfectly. This simplification should be kept in mind when analysing the surface-type specific fluxes.

Our results suggest that the emission from multiple surface types (Fig. 6 & 7) are very similar, and that some surface types contribute little to the footprint-average fluxes (Fig. S7-S8). This implies that a more detailed surface type characterization would have improved the model performance. The methods used for surface characterization hold promise for following

clearcut vegetation dynamics to address the vegetation recovery after site preparation and planting. More detailed vegetation classification was examined but found difficult as the vegetation after the clearcutting was sparse and the plant sizes were small. This caused the number of polygons for some vegetation classes in the training data to be very small. The vegetation growing on ditches had larger and more uniform surface area, and the classification of those would be easier than of individual saplings. The topography of the studied site is flat, which makes the classification between the ditch, tree and other vegetation types using the drone-derived elevation model to be more accurate. Here we could utilize precise georeferencing using Ground Control Point accurately measured in the open area of the site. For more detailed vegetation surveys, the resolution of the drone orthomosaic could still be increased to determine the leaf and branch structure of the smallest plants as the spectral differences are not only defined by species but also by e.g. plant health (Grybas and Congalton, 2021; Zhou et al., 2021). Parameters describing the structure, such as gray-level co-occurrence matrix, should be used additionally for classification. Alternatively, deep learning methods provide high classification accuracy by taking the structure into account without parametrisation (Onishi and Ise, 2021). Using the same sensors, increased resolution could be achieved by lowering the flight altitude resulting in increased flight time and battery capacity need. In addition, increased resolution can make generating training data and validating results more difficult as the number of segments increases and it is more difficult to decide which class the polygon represents, especially in sites like clearcut area with very detailed surface cover requiring multiple surface-type classes.

### 4.3 Clearcut peatland forests are net GHG sources

Despite the importance of peatland forests in the Nordic countries, little is known on the impacts of harvesting practices and alternative management chains on their GHG balance dynamics. In particular, GHG fluxes occurring shortly after clear-cutting and stand establishment have been rarely quantified (Mäkiranta et al., 2010; Korkiakoski et al., 2019, 2023). In this study, we employed the EC technique to quantify the $CO_2$, $CH_4$ and $N_2O$ fluxes from a recent clearcutting area, which was regenerated by planting after site preparation and ditch network maintenance. The findings demonstrate that our previously spruce-dominated fertile peatland forest was a major source of GHG emissions during the first full year (second growing season) after clearcutting and site preparation. The results indicate the $CO_2$ is the primary contributor to the annual GHG balance, accounting for 83% (23.3. t $CO_2$-eq ha$^{-1}$yr$^{-1}$) of the total global warming potential of the GHG emissions. It is also important to note that the $CO_2$ source strength of the clearcut area was *ca.* 10 t $CO_2$-eq ha$^{-1}$yr$^{-1}$ larger than NEE before the clearcutting (13.2 t $CO_2$-eq ha$^{-1}$yr$^{-1}$, Laurila et al., 2021). Our results are consistent with those previous studies on forested peatlands. A relatively similar fertile drained mixed forested peatland (Lettosuo) in southern Finland was found to be $CO_2$ neutral before harvesting as observed by EC measurements (Korkiakoski et al., 2023). After clearcutting and site preparation, the ecosystem turned into a strong $CO_2$ source, emitting initially 31 t $CO_2$-eq ha$^{-1}$yr$^{-1}$ but decreasing to 8.2 t $CO_2$-eq ha$^{-1}$yr$^{-1}$ six years after the harvest. This decrease was attributed to the increased $CO_2$ uptake by emerging vegetation and the concomitant decrease in $CO_2$ release from decomposing cutting residues decreased (Korkiakoski et al., 2023). At our site, the recovery of ground vegetation was observed as significant GPP (14.9 t $CO_2$-eq ha$^{-1}$ yr$^{-1}$) already at the second post-harvest growing season. This partially offset more than 35% of $R_{eco}$ which was mostly associated with $CO_2$ emissions. In a more southern

minerotrophic drained forested peatland (Tobo) in the Uppsala region of Sweden, $CO_2$ emissions were quantified using

chamber-based methods, with values ranging from 27 to 50 t $CO_2$-eq ha$^{-1}$yr$^{-1}$ in the second year following clearcut, depending on ditch management (Tong et al., 2022). Furthermore, Mäkiranta et al. (2010) reported chamber-based estimates of $CO_2$ emissions during the growing season from a clearcut drained oligotrophic peatland (Vesijako) located in southern Finland varied between 16 and 23 t $CO_2$-eq ha$^{-1}$ during the first three years after clearcutting.

The net $CO_2$ emissions from our study site Ränskälänkorpi clearcut area are comparable to EC-based measurements by Ahmed (2019; 20 t $CO_2$-eq ha$^{-1}$yr$^{-1}$) after clearcut of a fertile Norway spruce stand on mineral soil in Hyytiälä, Southern Finland. Furthermore, they are $20 - 30\%$ larger than the $CO_2$ emissions from 1-3 year old clearcuts on mineral soils in Southern and Central Sweden ( $16 - 18$ t $CO_2$-eq ha$^{-1}$yr$^{-1}$; Grelle et al., 2023). Kolari et al., (2004) observed smaller emissions (14 t $CO_2$-eq ha$^{-1}$yr$^{-1}$) 4 years after clearcutting an infertile Scots pine stand on mineral soil in Southern Finland. At Norunda,

Sweden, a clearcut former Norway spruce forest on mineral soil with shallow water table was identified as net source of $CO_2$ (NEE 16 t $CO_2$-eq ha$^{-1}$yr$^{-1}$) during the first and second year after harvest 16 and 11 t $CO_2$-eq ha$^{-1}$yr$^{-1}$, respectively. At this site, GPP and $R_{eco}$ exhibited fluctuations ranging between $5 - 14$ t $CO_2$-eq ha$^{-1}$yr$^{-1}$ and $21 - 23$ t $CO_2$-eq ha$^{-1}$yr$^{-1}$, respectively (Vestin et al., 2020).

The contribution of $N_2O$ and $CH_4$ emissions to the total annual GHG balance remained small despite their much higher global warming potential. Specifically, the contribution of $N_2O$ emissions was 17.6% (5.0 t $CO_2$-eq ha$^{-1}$yr$^{-1}$), while the $CH_4$ had only marginal importance (0.4%; 0.1 t $CO_2$-eq ha$^{-1}$yr$^{-1}$). The negligible share of $CH_4$ to net GHG emissions is in line with that found in Lettosuo and Tobo sites (Korkiakoski et al., 2019; Tong et al., 2022). Korkiakoski et al. (2019) estimated from chamber measurements that $N_2O$ emissions from Lettosuo site were 3.7 g $N_2O$ m$^{-2}$yr$^{-1}$ after clearcut, which makes more

than 11 t $CO_2$-eq ha$^{-1}$yr$^{-1}$. According to Tong et al. (2022), $N_2O$ emissions after clearcut at Tobo site contributed only 0.5-1.3% to total GHG emissions, likely due to the fact that the biweekly chamber sampling may have missed some of the high emission peaks and due to low soil moisture as the water table depth was low compared to similar studies. Note that the prior studies have utilized temporally and spatially discontinuous chamber measurements for observing $N_2O$ and $CH_4$ fluxes. Vestin et al., (2020) observed net $CH_4$ emissions between $0.3 - 1.5$ t $CO_2$-eq ha$^{-1}$yr$^{-1}$ and $N_2O$ emissions of $0.8 -$

1.1 t $CO_2$-eq ha$^{-1}$yr$^{-1}$ from the Norunda clearcut using flux-gradient approach. To the best of our knowledge, this study is the first to document EC-based $N_2O$ and $CH_4$ fluxes from a forest ecosystem after clearcutting.

Our results confirm earlier findings (e.g., Korkiakoski et al., 2023) that clearcutting increases the GHG emissions from boreal forested peatlands, at least in short term when compared to mature forests (Minkkinen et al., 2001; Ojanen et al., 2010; Alm

et al., 2023). To evaluate the climate effects of alternative harvesting methods (e.g. continuous cover forestry) in comparison to rotation forestry and clearcutting, the post-harvest dynamics of GHG emissions must be better known, calling for more and

longer follow-up studies (Korkiakoski et al., 2023). In Finland 390000 ha of fertile drained peatlands will soon be subject to choice between clearcutting and second even-aged rotation, or converting to other management regimes such as continuous cover forestry (Lehtonen et al., 2023) or partial rewetting. It is currently estimated that the conversion to continuous cover

forestry (which excludes clear-cutting but permits frequent heavy thinnings from above) could result in an annual reduction of the clearcut area in fertile peatlands by 16000 ha $yr^{-1}$ (Lehtonen et al., 2023). It is therefore evident that comparative long-term studies (but also modelling) between rotation forestry with clearcutting and alternative harvesting approaches across a spectrum of site characteristics are needed to facilitate the development of effective harvest management strategies to mitigate GHG emissions, especially those of $CO_2$ from peat decomposition, in boreal forested peatlands.

**5 Conclusions**

We measured $CO_2$, $CH_4$ and $N_2O$ fluxes the second growing season after clearcutting of a Norway spruce dominated boreal drained peatland forest in southern Finland using eddy covariance-based measurements. On the second growing season, the clearcut area was a significant source of GHG emissions with the annual total GHG balance dominated by the $CO_2$ emissions (23.3 t $CO_2$-eq $ha^{-1}yr^{-1}$, [22.4 − 24.1 t $CO_2$-eq $ha^{-1}yr^{-1}$ depending on the EC gap-filling method], 82.0% of total). The

765 $N_2O$ emissions (5.0 t $CO_2$-eq $ha^{-1}yr^{-1}$ [4.9 − 5.1 t $CO_2$-eq $ha^{-1}yr^{-1}$]) contributed 17.6 % while the role of $CH_4$ flux (0.1 t $CO_2$-eq $ha^{-1}yr^{-1}$ [0.1 − 0.1 t $CO_2$-eq $ha^{-1}yr^{-1}$], 0.4%) was negligible. We note that our study represents only a partial overview of a rapidly evolving forest ecosystem and that longer studies are needed to better understand the GHG budget of clearcut boreal peatland forests. However, the results presented herein reinforce the recently established understanding that clearcut peatland forests are significant source of GHGs.

Using the drone-based surface classification together with the EC measurements and statistical modelling, we estimated surface type specific $CH_4$ and $N_2O$ fluxes. The best-fitting models revealed that the highest $CH_4$ emissions in the study area originated from the plant-covered ditches and exposed peat, while the highest $N_2O$ emissions occurred from the exposed peat, litter, dead wood and living trees.

The role of exposed peat as a $CH_4$ and $N_2O$ emitter suggests the need for more detailed studies to understand the processes within this surface type. Based on this study, reducing the amount of exposed peat after clearcutting would be beneficial in decreasing the $CH_4$ and $N_2O$ emissions to the atmosphere. Similarly, reducing the litter input to the ground during harvesting would be beneficial in decreasing $N_2O$ emissions. The plant-covered ditches are important for the $CH_4$ emissions based on the

780 modelling and therefore an interesting future question is that how the $CH_4$ emissions change with respect to different plant species.

Finally, we note that our results are based on statistical modelling, and therefore recommend that manual chamber-based measurements be conducted in parallel with EC-based measurements to better constrain and validate surface-type specific flux estimates for better forest management of drained boreal forested peatlands.

## Data availability

The EC data will be made available upon manuscript acceptance. The data-analysis repository is available at GitHub https://github.com/LukeEcomod/FI-Ran_GHG_2022

## Author contribution

PS maintained the measurement infrastructure. OP processed eddy covariance data and QL, EMG and PS processed other data. PA and VT performed the UAV flights and developed the surface-type classification together with MP. OPT and JH developed the statistical models for GHG flux estimation. OPT, PA, JH, SL, AL, QL, EMG, OP, MP and RM analysed the statistical model results. OPT, PA, AL, SL, QL, EMG, OP, PS, VT and RM wrote the manuscript. All authors commented on the manuscript draft and gave permission to submission.

## Competing Interests

The authors declare that they have no conflict of interest.

## Acknowledgements

This study was funded by the Research Council of Finland (BiBiFe no.325680; PATCHEC 354298; Respeat 350184; Sompa 312932 and PREFER 348102), the European Union's (LIFE-IP CANEMURE-FINLAND project, grant number. LIFE17 IPC/FI/000002), the European Union's Horizon 2020 (HoliSoils project, grant agreement 101000289; Alfawetlands 101056844; Wethorizons 101056848, GreenFeedBack 101056921), and the Ministry of Agriculture and Forestry of Finland (TURNEE project, no. 4400T-2105). Authors also wish to acknowledge that this study was conducted with affiliation to the UNITE Flagship, which was funded by the Research Council of Finland (no. 337655). We also thank Anssi Liikanen and the field staff of the Natural Resources Institute Finland for their contribution by collecting the field data used in this study.

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
