# Peer review of "Eddy covariance fluxes of CO2, CH4 and N2O on a drained peatland forest after clearcutting"

_EGUsphere, 2024_

## Referee Comment (RC2)

**BG review of Tikkasalo et al**

**Overall:**
Scientific significance: excellent, scientific quality: excellent, Presentation quality: good
Overall this is an excellent paper. Many of my comments are requesting clarification or more details. Aside from these minor points, I have two major issues to point out:

The first about the lack of a spatially explicit flux modeling for CO2 as was done for CH4 and N2O. Much of the paper is justifiable building expectations for the impacts of spatial heterogeneity after clear cutting, and it is surprisingly absent in the results and discussion for CO2. In comparison to CH4 and N2O, I would expect CO2 to be easier to model given its strong relationships to variables already reported in the gap-filling discussion. The authors could take the GPP and respiration models used with gap-filling and apply the same spatial disaggregation technique as they did with CH4 and N2O.

The second issue is about the methane flux results. The flux estimates from the plant-covered ditch surface-type are extremely large, almost unbelievably so. These results need to be justified and put in context of other methane emissions. Given that the areal contributions of this surface type and therefore their weights within the footprint, are so small, it could be very difficult to have confidence in these results. In addition to comparisons to chamber fluxes or other studies, I would suggest investigating the robustness of the methane surface-type model with a simulation.  Generate a flux for each surface-type based on your equations 3 and 4, calculate the theoretical EC observation after multiplying by the pixel footprint weight and summing, then add some reasonable random noise. Then apply your dissaggregation model and see if you can recover the original parameters you used to generate the fluxes. This is a straightforward way to test whether your dataset is under-determined or not. If you do not have enough variability in footprints weights from surface-types to recover your simulated fluxes, then you will have to reduce the complexity of surface-types or use a longer time series of data.

**General comments:**
- In figure 1 and throughout when color-coded landcover types are displayed: It is difficult to distinguish similar colors. The greens in particular all look the same. A more divergent color scheme would improve readability throughout the paper.

- Model predicative performance for the gap-filling ML models and the spatially explicit footprint flux models is evaluated and reported using R-squared. Whenever R-squared is reported, the slope and intercept of the regression should also be reported. R-squared describes the variance around the fit, but the slope and intercept describe model bias which is equally important. I also suggest providing the RMSE as a more useful metric than R2 because it is in comparable units.
- Section 2.8:
    - The methods described for surface-type modeling are the same as those used by Ludwig et al. 2024 from your introduction, and it should be cited here as well.
    - Can you please provide some justification for your choice of prior distributions.
    - Please describe your tests for convergence and their outcomes.
    - Please clarify that only non-gap-filled data were used in the surface-type modeling analysis
    - Why use LOO cross validation for the surface type modeling, when you already have withheld data in artificial gaps created for the gap-filling ML models?
- Figure 4 and 5: include slope and intercept on the fit depicted in panel c.
- Figure 6: The bold line for the central quartile is hard to distinguish, can you make it bigger?
- Table 3: I understand that the gap-filled budgets in the second and third column are agnostic to the area and make-up of the footprint. How are the surface type modeled fluxes summarized to comparable numbers to the gap-filled EC data, given that each observation has a different distribution and weight of surface types? The modeled fluxes can be weighted by footprints before summarizing to a budget, but due to gaps, there are timepoints without footprints. It would make more sense to me to use your surface-type models to calculate the budgets for the entire domain in your Figure 1, and then similarly apply the gap-filled time series of fluxes to the same area when summarizing, rather than reporting on a per area (ha-1) basis. By controlling the areal extent of this comparison it might also reveal interesting agreements or discrepancies between the surface-type model budgets and the footprint-agnostic gap-filled budgets.
- Section 4.1 first paragraph:

- The spatial heterogeneity is generally put in context of similar ecosystems and other clear-cutting studies. But what is lacking is a quantitative comparison of the magnitude of these fluxes determined here (figure 7) to other studies. For example, is your exposed peat flux typical of peat ch4 fluxes? While I am not surprised by a slight uptake of methane in some surface types, it is surprising to see methane uptake in the ditch surface water. Similar features in polygonal tundra are large methane sources. The methane flux from plant covered ditches, the vast majority of all methane at this site, is alarmingly large, as in, it is similar to methane fluxes measured by eddy covariance at active landfills in warm climates. This result needs to be put in context of other fluxes and justified.
- In table 2, you set up an investigation of scenarios to determine the level of complexity to use in the spatial disaggregation of fluxes. This is a great tool for supporting the robustness of your surface-type model results. You present results from the best model of the set described in the table. I would like to see more results on all scenarios. Specifically, how do the surface type flux estimates change in each version in table 2? In two of the five versions, your highest flux type is lumped with your lowest flux type, and discussing how the fluxes turn out in these scenarios would help provide confidence in the model results.
- Figures S7 and S8: Please include the panel letter designations in the legend as well for clarity.

**Specific comments:**
Line 88: Need space after period at the end of sentence
Line 107: Missing word. "[The] likely reason for this...
Line 247: missing space in citation for (Kljun et al 2015)
Line 565: Should cite Ludwig et al. 2024 here as well.
Line 581: Typo 'emissionsdd,'

---

## Author Comment (AC1)

**Response to comments by Anonymous Referee #1**
The author comments and answer are written without highlighting, while the comments of the Anonymous Referee #1 are highlighted in *cursive*.

*The manuscript presents results of the total GHG balance (CO2, N2O and CH4) from a clearcut stand on a fertile peatland in Finland. The manuscript uses one-year measurements of eddy covariance to quantify the strength of source from clearcutting. Combining results with a UAV-based land classification and statistical modelling the authors split the source of fluxes per land class (i.e., surface-type).*
*My overall assessment of the project's objectives and approach is that this is very important and interesting work, especially when it addresses the full GHG balance which current literature fails to address adequately. However, I have some concerns/comments/suggestions regarding the methodology and the approach the authors took in this study. I will aim to first discuss my main concerns/comments.*

**Answer:** We thank Referee #1 for the constructive comments that have greatly improved the manuscript. Please see below our specific answers to comments. The line numbers below refer to the revised version of the manuscript with track changes

1. *The authors claim that this study aims to investigate the impact of clearcutting on the GHG balance of forested peatlands. Yet in lines 136-138, they state that "stand regeneration was carried out in summer 2021 through ditch mounding and planting of Norway spruce seedlings". So:*

    1.1 *This is no longer a "clearcut" site since it has been replanted. It is a restock site on its second growing season (as the authors have stated multiple times throughout the manuscript) and hence the strength of source is no longer reflective of a clearcut practice (due to GPP).*

    **Answer:** We agree on this, text was modified just above the aims to make it clear that this paper deals with 2nd year measurements of GHG emissions. Clear-cutting, ditch mounding and replanting are common practices to establish 2nd tree generation when applying even-aged rotation forestry on drained peatland forests. The forestry measures conducted at our study site are thus common and representative for even-aged forestry. We agree our terminology was misleading, and we did not investigate impact of clearcutting but documented GHG fluxes (and GHG balance in terms of $CO_2$-eqv.) over 2nd post-clearcut year. Investigating the impact of clear-cutting would have required flux measurement from a reference period before clear-cutting. This is hopefully now corrected in the revised manuscript.

    1.2 *Ditch mounding was used before planting, which suggests to me that the site was disturbed prior to measurements and hence not again representative of a clearcut site. In fact, if indeed any ditch mounding was applied after clearcutting, it means that the land classification reported is also not representative of the post-felling fluxes.*

**Answer:** Ditch mounding is a common practice conducted on drained peatlands after clear-cutting to improve seedling survival. Both clear-cutting with heavy forest machines and ditch mounding create disturbance to peat soil surface, which is reflected in surface type proportions and their subsequent dynamics during the first years after the disturbance. Our surface type classification is done in the summer of 2022 which is the same year the EC measurements reported in this manuscript were performed. We have added clarification in Section 2.6. that the surface type classification is based on the drone imaging which were captured in June 2022.

**1.3** *The authors mention that this is a fertile peatland, however, they didn't give us any further information as to how they are fertile. Was the site historically fertilised prior to planting or is because of a natural fertilisation over a number of rotations? I believe an international audience would like to know a little more information about the particulars of Finnish peatlands.*

**Answer:** Thanks for the comment. We have elaborated text in this regard under material and methods section, and now we provide more information on site fertility type. At present the Ränskälänkorpi research site is well-drained, Norway spruce dominated and represents mainly nutrient-rich Herb-rich (Rhtkg II) and *Vaccinium myrtillus* (Mtkg II) site types drained peatland forest (Laine et al., 2012).

2   *Fluxes presented are from a single year. I understand that authors may feel compelled to present their very interesting work as soon as the first results are available, however, it is very rare, if not I dare say totally unrealistic, to draw any conclusions on the source/sink of a site with simply a single year especially when this year is not also representative of the actual effect of the forest management practice the study claims (see point 1). There is still a huge gap in our knowledge of what is the initial pulse of GHG immediately after clearcutting, and I believe the authors may have missed the opportunity here to capture a potentially significant contribution from the first few months and prior to any planting or mounding.*

**Answer:** We agree with the referee that we have missed a potentially important emission contribution from the first growing season following the clearcutting. Our data presents a snapshot of continuously evolving forest patch roughly a year following the clearcut. However, we feel that it is important to report also these snapshots from rapidly changing ecosystem especially as one of our target is to characterize which surface types are impact the most to $CH_4$ and $N_2O$ emissions.

We have reformulated the conclusions of the study that hopefully also reflect the fact that the temporal length of our study is limited.

*3    The modelling, although very interesting, I don't believe it has worked as expected particularly for methane. I believe the fact water table depth (WTD) or even soil moisture (theta) was ignored in the modelling was a major overlook since we know (and as the authors themselves demonstrated with Figure 3) both fluxes but particularly CH4 are strongly correlated. Furthermore, another pitfall was the choice of Tair over Tsoil. Volumetric heat capacity changes linearly with moisture, so for wet peatlands I would expect changes in Tsoil to have a bigger impact that those Tair. So, potentially, there was an underestimation of the flux and hence lower strength in the model. Finally, I believe the exclusion of some surface-types from the CH4 model may have resulted in reduced model efficiency, as it clearly worked for N2O. The authors claim that CH4 emissions were not surface dependent (lines 558-559), however, from a work at a Scottish peatland restoration site (Mazzola et al. 2021, European Journal of Soil Science) it was found that CH4 fluxes were significantly different with micro-topography, including water pools. Not considering any interaction of flux with water or moisture it is likely to result to a mismatch between model and data.*

**Answer:** We thank the referee for suggesting to add soil water availability describing variable to the models. We tested both the water table depth (WTD) and soil moisture, $\theta$, as well as models where the surface type contribution of temperature was included or removed (the term with $\delta$ in the model equations). When these new models and old models from the first submission of the manuscript were compared the best model for both compounds was found to be the one with $\theta$, 9 surface types and the $\delta$ temperature term included. Since the WTD and $\theta$ are similar metrics we do not report the modelling results for the WTD models but state that the best model was the one based on $\theta$.

Because of the reasoning that we do not have measurements what is the water availability at different locations of the studied area we only added a general $\theta$ dependency term in the new model (Eq. 4 in the revised version of the manuscript). We have adjusted the text in the revised version of the manuscript where needed to match the new best models. We have removed figures S7 and S8 from the first submission as presenting the flux estimate for each surface type as a function of soil moisture and air temperature was challenging. The inclusion of $\theta$ in the model decreased the surface type specific fluxes (fig. 7) since part of the emissions are now attributed to the water availability term.

We have added also the reference to Mazzola et al., (2021) in the discussion section in lines 664-665.

"Also Mazzola et al., (2021) found, based on chamber measurements, that there was a clear difference between surface type specific $CH_4$ emissions on a restored bog site in northern Scotland"

We also tested that using $T_{soil}$ instead of $T_{air}$ would lead to slight improvement of the best model with $N_2O$ but not for $CH_4$. Thus, we opted to keep $T_{air}$ as the independent variable because of the reasoning above that it is likely more similar across the surface types than $T_{soil}$ that is only measured at three locations. We have added recommendation based on this results to the methodological outlook section in the discussion on lines 673-675:

"For the best models we also tested replacing $T_{air}$ with mean soil temperature measured at the tree locations shown in Fig. 1. For $N_2O$ this produced slightly better fit in terms of ELPD-LOO (difference of 74 units). This suggests that especially for understanding $N_2O$ emissions, measuring the surface type specific soil temperature would be beneficial."

4   *I believe the uncertainty presented in Table 3 for CH4 and N2O re-enforce my opinion that the model for CH4 did not perform well (uncertainty mismatch) comparing to N2O (EC uncertainty within modelled).*

**Answer:** The new median model prediction for $CH_4$ in Table 3 is closer to the EC derived estimate and also the 95% HDI range is slightly lower. The addition of soil moisture to the model as suggested by the referee has increased the performance of $CH_4$ in particular.

5   *I am also surprised that N2O fluxes were not high after clearcutting. Yamulki et al. 2021 (Biogeosciences) found high N2O on an organo-mineral (30-60cm peat layer over a mineral layer). With a high fertility peatland when trees removed and WTD increases I would expect pulses of N2O. The authors demonstrated that the model was unable to capture the pulse of N2O in August. Was that pulse close to a rainfall event? If so, ignoring relationship WTD and/or theta, hindered the model's predictive capability.*

**Answer:** The period with high $N_2O$ emissions lasted approximately 10 to 15 days in late July and early August and there indeed was a relatively strong precipitation event (33.4 mm of rain during July 23) slightly before the high $N_2O$ emissions. The precipitation event increased soil moisture and it started to decline after the precipitation event (see Figure below). As a response to this, we modified our $N_2O$ flux model by including a common term describing $N_2O$ flux response to soil moisture as suggested by the referee, unfortunately even with this addition the model was not able to capture the peak in $N_2O$ emissions.

[Figure]

**Figure 1: Coincidence of N$_2$O flux with precipitation events**. N$_2$O flux (top plot), soil moisture (bottom plot, continuous line) and precipitation (bottom plot, bars) time series around the period with high N2O emissions. The approximate beginning and end of the high N2O emission period are highlighted with red dashed lines.

6    *I also found very difficult to evaluate the strength of the model's accuracy. R-squared and RMSE although they give some indication of the model's predictive capabilities, it was difficult to evaluate further the model, especially where little explanation was given for the LOO statistic. I understand this is a MC-based modelling approach, but I wonder whether a statistic like Akaike Information Criterion, or a significance level for the slope and intercept of the model vs data would be very useful to evaluate the explanatory capacity of the model.*

**Answer:** We have changed the model evaluation in the revised version of the manuscript. We rank the models whose parameters have been estimated with the MCMC technique using only the ELPD-LOO metric and show the performance of the best models in Fig. 4 and 5 with R$^2$, RMSE and also report the slope and intercept of a linear fit between the (MAP) estimated and measured flux. Additionally, we have added text how to interpret the ELPD-LOO on lines 394-396:

"The compare function ranks the models based on the expected log posterior density of the left out samples. While a single ELPD-LOO value is not easy to interpret in terms of model performance, models are straightforward to compare against each other as higher value of ELPD-LOO marks better performance."

7   *Surface-specific splitting on fluxes were performed only for CH4 and N2O, however, CO2 was ignored. Why was that? I believe it would have been a great opportunity to repeat the process for CO2.*

**Answer:** We considered doing similar surface-specific analysis with $CO_2$ flux observations but opted not to do so due to the following reasons:
1) the vegetation was rapidly recovering from the clearcut during the growing season and it is unclear how to take this recovery into account in this kind of analysis since the responses to $CO_2$ flux drivers change rapidly in time. For instance, due to the recovery the ecosystem $CO_2$ flux response to radiation was rapidly changing during the growing season. We could follow e.g., Buzacott et al., (2024) and assume certain kind of seasonal patterns for the parameters describing gross primary productivity light response curves and ecosystem respiration temperature dependence, however it is unclear what kind of seasonality would be appropriate in this recovering ecosystem. Moreover, this seasonality is likely different for different surface types resulting in many fitted parameters and hence large uncertainty.
2) our map delineating the clearcut surface into different categories is static, i.e. it does not vary in time, however pioneer species were spreading in the clearcut area during the growing season. This should be considered if the surface-specific were to be derived from $CO_2$ flux observations. Due to these reasons we opted to report only the ecosystem-scale $CO_2$ observations without trying to disaggregate the $CO_2$ fluxes to different surfaces.

8   *I would have liked to see more of an investigation not only how much of the flux is coming from each soil type, but what are the underlying processes by discussing correlation between vegetation, flux and climatological variables and topography.*

**Answer:** Thanks for the suggestion, we have done small additions to discussion about mechanisms of fluxes from surface types were added in 4.1 but at the same time, we want to avoid adding too much specific details about the underlying processes as our results would greatly benefit from comparison against chamber measurements.

9   *The manuscript presents the results of a footprint analysis, followed by a discussion on its potential limitations. It was unclear to me how the footprint was used in further analysis. More importantly, the manuscript is unclear whether footprint was used to either calculate the total area of for surface-type classification of even whether the fluxes were adjusted for footprint contribution once they have been split into different surface-type. This can have a potential major implication on how results are interpreted. It is expected, surface-types closer to the eddy covariance tower to have greater contribution. If for example, plant filled ditches are closer to the tower then potentially their contribution will be larger. Ignoring the combined effect of the surface-type distribution across the area can lead to bias. I suggest the authors review the methodology followed by Budishchev et al. 2014 (Biogeoscience) and revisit some of their approaches.*

**Answer:** Thank you for this comment. The footprints were utilized when deriving the surface-specific emissions based on the EC data, please see manuscript Eqs. (5-8), specifically the term $\varphi_{i,j}$ in the equation. This way the models were able to account for the heterogeneity of the clearcut surface and the model could be used e.g. to estimate

surface-specific fluxes, see manuscript Fig. 7. The referee is right that the coverage of different surfaces within the EC footprint may depart from the share of those surfaces in the whole clearcut area and hence in such cases EC is observing a biased sample of the clearcut-atmosphere exchange (see also Chu et al., 2021). In response to this comment, we added a column in Table 1 where we report the mean share of each surface type in the EC footprint and compare those against their share of the overall clearcut surface. Note that the modelled flux estimates in Table 3 were already calculated so that they represent the whole clearcut surface and not the EC footprint. This was achieved by utilizing their share of the overall clearcut surface in Eq. (7) when using the fitted models for estimating the fluxes. We agree that this was not clearly articulated in the manuscript and hence tried to clarify this in Table 3 caption and by adding text on lines 548-660.

10 *The manuscript presents a section on footprint analysis and considerations with a discussion element. However, it was not clear to me how the footprint was used other than simply for presentation purposes. Was the footprint used for the classification of the surface-type?*

**Answer:** We tried to clarify the usage of footprints in our previous answer. The footprints were not used in the classification of the surface into different classes, but this was done independently with a combination of drone imaging (Sect. 2.5) and machine learning algorithms (Sect. 2.6). We added text clarifying this on manuscript line 271.

11 *Following the point from above, it wasn't clear whether the surface-type classification was for the whole of the clearcut area or for the footprint. This is potentially key to interpreting the results. Land within the footprint of the tower would have bigger contribution*

**Answer**: As shown in manuscript Fig. 1, the whole clearcut surface was classified into different surface categories and this was done independently from footprint analyses. We then overlaid footprints on this map with surface classes to evaluate how much different surface categories were contributing to the EC observations. This information was then in turn used in developing the model (manuscript Eqs. 3-8) which allowed us to evaluate surface-specific emissions. See our response above for the EC footprint sampling bias.

12 *Lines 649-656, the CO2 emissions from the peatland are compared to mineral soil. The authors must understand matching fluxes in these two different soil types does not equate validity of measurements due to underlying differences in carbon stocks and respiratory processes.*

**Answer:** We are aware that the fluxes between different surface types cannot be used to validate measurements. In the discussion section 4.3 our aim is to put our measurements into context of other post-clearcut young boreal and hemiboreal stands.

*Some further comments:*
1. *The introduction only lightly touches on the importance of N2O and the current gap in knowledge.*

**Answer:**  Thank you for this comment. We have included new information on the challenges of measuring $N_2O$ fluxes and the importance of accurately estimating them in relation to their contribution to the global GHG budget (see lines 90-94)

2. *The introduction also did not make clear what is the uniqueness of this study. In my opinion, this is a novel approach which aims to close the total GHG balance for the boreal and specifically the Fennoscandia, but it was not explicitly highlighted.*

   **Answer:** We have refined the last paragraph of the introduction to present why our study is needed.

3. *Figure 3 presents a correlation analysis. Are these correlations statistically significant? It was not discussed what the correlations mean for the underlying processes. Keeping the current discussion, I propose this analysis is removed. Alternatively, it can be significantly reduced to include key significant correlations which may further used in the discussion to understand processes.*

   **Answer:**  The presented correlations are statistically significant. After consideration we decided to keep Fig. 3 in the manuscript, even though it could also be moved to supplement. The reasoning for our choice is, that with the revised version the supplement is already quite heavy and we want readers to be able to find the GHG flux correlations from the main text easily as this is something we except potential readers to be interested about. We have added a note to the start of section 2.7. that clarifies that in this study the correlation analysis is only used as a basis for selecting environmental variables for statistical flux modelling.

4. *Having said that, the manuscript has a lengthy discussion on the modelling. Although, important to highlight modelling limitation and potential pitfalls, I felt there was a little less time spend discussing the underlying processes that are related to different surface-types.*

   **Answer:** Please see our answer to comment 8.

5. *Figure 6 was very difficult to understand. The points and bars where too small for some variables and hence difficult to convey the message. I wonder whether there is an improved way to present the parameter values. A line across the zero would also have been helpful.*

   **Answer:** We have added a line across the zero and increased fonts, marker sizes and line widths for Fig. 6 to improve readability.

6. *I don't understand since we have the parameter values and range in Figure 6, why we had to set the surface-type contribution to one, to "visualise" the parameters in Figure 7. Why not simply present with the estimate surface-type contribution percentage what is the total flux from each and the percentage of the total flux measured by the eddy covariance tower? I believe this information is far more useful and citable for future*

*work.*

**Answer:** We agree with the referee that presenting the contribution of a surface type to the overall flux would be the most informative way of communicating our results. However, since we wanted to work with more normal distributed data we needed to take the log-transform of the measured flux value. Because of this transform once one takes the back transformation (Eq. 9) what is left is a multiplicative model. For multiplicative models it is challenging to determine a rule for calculating a contribution of a single surface type to the overall flux. Furthermore, none of these rules would be such that the contributions would sum to unity.

We decided to go with the current presentation where we show 1) the distribution of the estimated parameters 2) surface type specific flux distribution by assuming unity surface coverage for each surface type in turn 3) scenario-based calculations how adding a single surface type influences the estimated flux value (Figs. S7-S8)

7. *It was interesting that the study found N2O emission during snow cover. This is potentially a important find which the manuscript did not discussed in its full extend. Of course, the single year worth of data makes it very difficult but even so, it is important to highlight its importance and whether something similar has been reported before.*

**Answer:** Thank you for this valuable suggestion. We have added a brief discussion of the relevance of $N_2O$ fluxes during the snow-covered period to the annual budget, as well as the implications of winters that are not as normal as the one studied, in the revised version of the manuscript. See lines 601-606 for further details.

8. *The conclusion sections is a repetition of information already given in either the abstract or the results section. The section requires a refocus to really provide a concluding message from the study.*

**Answer:** We have refined the conclusion section in the revised version of the manuscript.

**References**
Buzacott, A. J. V., van den Berg, M., Kruijt, B., Pijlman, J., Fritz, C., Wintjen, P., and van der Velde, Y.: A Bayesian inference approach to determine experimental *Typha latifolia* paludiculture greenhouse gas exchange measured with eddy covariance, Agric. For. Meteorol., 356, 110179, https://doi.org/10.1016/j.agrformet.2024.110179, 2024.

Chu, H., Luo, X., Ouyang, Z., Chan, W. S., Dengel, S., Biraud, S. C., Torn, M. S., Metzger, S., Kumar, J., Arain, M. A., Arkebauer, T. J., Baldocchi, D., Bernacchi, C., Billesbach, D., Black, T. A., Blanken, P. D., Bohrer, G., Bracho, R., Brown, S., Brunsell, N. A., Chen, J., Chen, X., Clark, K., Desai, A. R., Duman, T., Durden, D., Fares, S., Forbrich, I., Gamon, J. A., Gough, C. M., Griffis, T., Helbig, M., Hollinger, D., Humphreys, E., Ikawa, H., Iwata, H., Ju, Y., Knowles, J. F., Knox, S. H., Kobayashi, H., Kolb, T., Law, B., Lee, X., Litvak, M., Liu, H., Munger, J. W., Noormets, A., Novick, K., Oberbauer, S. F., Oechel, W., Oikawa, P., Papuga, S. A., Pendall, E.,

Prajapati, P., Prueger, J., Quinton, W. L., Richardson, A. D., Russell, E. S., Scott, R. L., Starr, G., Staebler, R., Stoy, P. C., Stuart-Haëntjens, E., Sonnentag, O., Sullivan, R. C., Suyker, A., Ueyama, M., Vargas, R., Wood, J. D., and Zona, D.: Representativeness of Eddy-Covariance flux footprints for areas surrounding AmeriFlux sites, Agric. For. Meteorol., 301–302, 108350, https://doi.org/10.1016/j.agrformet.2021.108350, 2021.

Laine, J., Vasander, H., Hotanen, J.-P., Nousiainen, H., Saarinen, M., and Penttilä, T.: Suotyypit ja turvekankaat-opas kasvupaikkojen tunnistamiseen, 2012.

Mazzola, V., Perks, M. P., Smith, J., Yeluripati, J., and Xenakis, G.: Seasonal patterns of greenhouse gas emissions from a forest-to-bog restored site in northern Scotland: Influence of microtopography and vegetation on carbon dioxide and methane dynamics, European Journal of Soil Science, 72, 1332–1353, https://doi.org/10.1111/ejss.13050, 2021.

---

## Author Comment (AC2)

**Response to comments by Anonymous Referee #2**
The author comments and answer are written without highlighting, while the comments of the Anonymous Referee #2 are highlighted in *cursive*.

*Overall:*
*Scientific significance: excellent, scientific quality: excellent, Presentation quality: good*
*Overall this is an excellent paper.*

**Answer:** We thank Referee #2 for the productive feedback on our study that has greatly improved the study. Please see our specific answer to the comments below. Note that we have added numbers to the comments. The line numbers below refer to the revised version of the manuscript with the track changes.

*Many of my comments are requesting clarification or more details. Aside from these minor points, I have two major issues to point out:*

*1. The first about the lack of a spatially explicit flux modeling for CO2 as was done for CH4 and N2O. Much of the paper is justifiable building expectations for the impacts of spatial heterogeneity after clear cutting, and it is surprisingly absent in the results and discussion for CO2. In comparison to CH4 and N2O, I would expect CO2 to be easier to model given its strong relationships to variables already reported in the gap-filling discussion. The authors could take the GPP and respiration models used with gap-filling and apply the same spatial disaggregation technique as they did with CH4 and N2O.*

**Answer:** We considered analysing $CO_2$ flux observations similarly as done for $CH_4$ and $N_2O$, but opted not to. Please see our response to Referee #1 comment 7 for the reasons for this decision.

*2. The second issue is about the methane flux results. The flux estimates from the plant-covered ditch surface-type are extremely large, almost unbelievably so. These results need to be justified and put in context of other methane emissions. Given that the areal contributions of this surface type and therefore their weights within the footprint, are so small, it could be very difficult to have confidence in these results. In addition to comparisons to chamber fluxes or other studies, I would suggest investigating the robustness of the methane surface-type model with a simulation. Generate a flux for each surface-type based on your equations 3 and 4, calculate the theoretical EC observation after multiplying by the pixel footprint weight and summing, then add some reasonable random noise. Then apply your dissaggregation model and see if you can recover the original parameters you used to generate the fluxes. This is a straightforward way to test whether your dataset is under-determined or not. If you do not have enough variability in footprints weights from surface-types to recover your simulated fluxes, then you will have to reduce the complexity of surface-types or use a longer time series of data.*

**Answer:** We thank the referee for this suggestion. We ran the suggested test such that one author calculated "an artificial flux data set" as suggested and then another author performed the model parameter estimation without knowledge of what were the correct parameter

values. The correct parameters along with MAP estimates from the parameter estimation are shown in Table 1 below. Since the author performing the model estimation did not know how many surface types the correct answer set had, he ran the parameter estimation with 3,4,5,6 and 9 surface types. The best performing set was ST6 closely followed by ST9. From Table 1 it can be seen that since dead wood and harvest residue and field layer and living trees have the same correct $\gamma$ and $\delta$ the correct number of surface types was six in the artificial flux data set.

**Table 1**: Correct parameter values (columns 2-5) and the maximum a posterior (MAP) estimates of the same parameters (columns 6-9) for the artificial data set and parameter estimation performed with it. In columns 6-9 two values are given for each parameter: the first is from the parameter estimation using six surface types and the latter from using nine surface types.

| Surface type | $\alpha$ | $\beta$ | $\gamma$ | $\delta$ | $\alpha$ (MAP) ST6 / ST9 | $\beta$ (MAP) ST6 / ST9 | $\gamma$ (MAP) ST6 / ST9 | $\delta$ (MAP) ST6 / ST9 |
|---|---|---|---|---|---|---|---|---|
| - | 2.0 | 0.0 | | | 2.1 / 2.1 | $1.4 \cdot 10^{-5}$ / $6.9 \cdot 10^{-5}$ | | |
| Dead wood | | | -0.1 | 0.0 | | | -0.12 / -0.12 | $3.9 \cdot 10^{-4}$ / 0.0017 |
| Harvest residue | | | -0.1 | 0.0 | | | -0.12 / 0.30 | $3.9 \cdot 10^{-4}$ / 0.0011 |
| Exposed peat | | | 0.1 | 0.3 | | | 0.10 / 0.13 | 0.40 / 0.40 |
| Litter | | | 0.3 | 0.001 | | | 0.34 / 0.20 | 0.003 / 0.004 |
| Bottom layer (mosses) | | | 0.0 | 0.0 | | | NA / -0.9 | NA / 0.24 |
| Field layer | | | -0.2 | 0.0 | | | -0.25 / -0.27 | $2.4 \cdot 10^{-5}$ / $1.4 \cdot 10^{-5}$ |
| Living tree | | | -0.2 | 0.0 | | | -0.25 / -0.35 | $2.4 \cdot 10^{-5}$ / 0.0035 |
| Plant covered ditch | | | 5 | 0.0 | | | 6.31 / 6.2 | 0.014 / 0.0038 |
| Ditch (water surface) | | | -2 | 1.8 | | | -2.61 / -2.7 | 2.26 / 2.28 |

Additionally, below is a figure showing the distribution of the estimated parameters

[Figure]

**Figure 2**: Inferred parameters from the artificial data set, their distribution and correct parameter values. Subfigures a-b show the estimated and correct parameters for the model with six surface types and c-d for the model with nine surface types.

In our opinion, the test showed that there is enough variability in the footprints that the modeling can be performed with the experimental data set at hand. However, it is also clear that the estimates for the both ditch types can be biased (e.g., 23%-26% for the plant covered ditch in this test).

The methane fluxes for individual surface types decreased in the revised version of the model since some of the flux is now attributed to the term with soil moisture. We are stating already that the method to calculate surface type specific fluxes is an extrapolation of the model (line 510). We feel, however, that the publication of these extrapolations is needed for later comparison against e.g., chamber measurements.

*General comments:*

*3. In figure 1 and throughout when color-coded landcover types are displayed: It is difficult to distinguish similar colors. The greens in particular all look the same. A more divergent color scheme would improve readability throughout the paper.*

**Answer:** We have adjusted the colors in the revised version of the manuscript.

*4. Model predicative performance for the gap-filling ML models and the spatially explicit footprint flux models is evaluated and reported using R-squared. Whenever R-squared is reported, the slope and intercept of the regression should also be reported. R-squared describes the variance around the fit, but the slope and intercept describe model bias which is equally important. I also suggest providing the RMSE as a more useful metric than R2 because it is in comparable units.*

**Answer:** We have added slope and intercept information to the flux model and gap filling ML-model comparison. The best model selection is done solely based on the ELPD-LOO in the revised version.

*5. Section 2.8:*
*The methods described for surface-type modeling are the same as those used by Ludwig et al. 2024 from your introduction, and it should be cited here as well.*

**Answer:** We now cite Ludwig et al., (2024) in section 2.8.

*6. Can you please provide some justification for your choice of prior distributions.* ▪
*Please describe your tests for convergence and their outcomes.* ▪
*Please clarify that only non-gap-filled data were used in the surface-type modeling analysis*

**Answer:** Our decisions on choosing prior distributions was to keep the priors as uninformative as possibly while still incorporating the little knowledge that we have of the system. The addition of $\theta$ (soil moisture) to the model makes the interpretation of the parameters slightly more challenging i.e., we cannot say anymore that the variable $\alpha$ is the base gas emission rate at $T_{air} = 10°C$. For this reason we went with the normally distributed priors around zero mean for both $\alpha$ and $\gamma$ and $\zeta$. We also briefly considered using uniform priors but neglected this option as it would've meant that we believe that high values of these parameters are as likely as those near zero. For the temperature response parameters ($\beta$ and $\delta$) we went with exponential distributions as we assume that above $T_{air} = 10°C$ the effect of temperature to emissions is positive. Lastly, we ensured that the values we chose for the standard deviation and rate parameters of the priors were such that the full width at half maximum (FWHM) of the prior predictive distributions is at least two times of FWHM of the observations.

The convergence checks are run by default in the PyMC sampler. Most important for us is the Gelman-Rubin statistic (r-hat). The sampler warns if r-hat is higher than 1.01 for any parameter (the source code for the convergence checks can be found in the PyMC repository https://github.com/pymc-devs/pymc/blob/main/pymc/stats/convergence.py).

Only non-gap-filled data were used in the surface type modelling analysis.

*7. Why use LOO cross validation for the surface type modeling, when you already*

*have withheld data in artificial gaps created for the gap-filling ML models?*

**Answer:** We use the whole available gas flux data sets to fit the surface type models. This means that the artificial gaps that were created in developing the gap-filling ML models are not present when we develop the surface type models. We have added clarification to lines 387-389 in the revised version of the manuscript.

"The full, non gap-filled, EC flux data sets were used in the parameter estimation i.e., the artificial gaps introduced to the flux data sets for developing the gap-filling model were not present in this parameter estimation."

8. *Figure 4 and 5: include slope and intercept on the fit depicted in panel c.*

**Answer:** This information has been added to the revised version of the manuscript.

9. *Figure 6: The bold line for the central quartile is hard to distinguish, can you make it bigger?*

**Answer:** We have increased the line width for the central quartiles. We have also made several other changes to Fig. 6 to improve readability as suggested by Referee #1.

10. *Table 3: I understand that the gap-filled budgets in the second and third column are agnostic to the area and make-up of the footprint. How are the surface type modeled fluxes summarized to comparable numbers to the gap-filled EC data, given that each observation has a different distribution and weight of surface types? The modeled fluxes can be weighted by footprints before summarizing to a budget, but due to gaps, there are timepoints without footprints. It would make more sense to me to use your surface-type models to calculate the budgets for the entire domain in your Figure 1, and then similarly apply the gap-filled time series of fluxes to the same area when summarizing, rather than reporting on a per area (ha-1) basis. By controlling the areal extent of this comparison it might also reveal interesting agreements or discrepancies between the surface-type model budgets and the footprint-agnostic gap-filled budgets.*

**Answer:** This information was missing from the previous version of the manuscript. In the revised caption for Table 3 we are stating that the modelling approach estimate is calculated with the share of each surface type from the whole clearcut area not from individual footprints. If we understood correctly what the referee is asking, the revised Table 3 has the data that is suggested here. The per area fluxes can be converted to the whole clearcut area flux by multiplying with the clearcut area (ca. 6.1 ha).

11. *Section 4.1 first paragraph:*

*The spatial heterogeneity is generally put in context of similar ecosystems and other clear-cutting studies. But what is lacking is a quantitative comparison of the magnitude of these fluxes determined here (figure 7) to other studies. For example, is your exposed peat flux typical of peat ch4 fluxes? While I am not*

*surprised by a slight uptake of methane in some surface types, it is surprising to see methane uptake in the ditch surface water. Similar features in polygonal tundra are large methane sources. The methane flux from plant covered ditches, the vast majority of all methane at this site, is alarmingly large, as in, it is similar to methane fluxes measured by eddy covariance at active landfills in warm climates. This result needs to be put in context of other fluxes and justified.*

**Answer:** The fluxes from different surfaces shown in Figure 7 are different from measured results. Open water ditch should be large $CH_4$ source but it's not reflected in our results. One reason is the main ditch which contribute most of $CH_4$ emissions in our site was identified as plant covered because of vascular plants growing near the ditch. This also explained the large emissions from plant covered ditch we got.  The $CH_4$ fluxes from exposed peat varied in our study site based on our chamber measurements, depending on the water table in the location. It's difficult to quantitatively compare Figure 7 with measured results, because they are calculated by setting specific surface-type contribution to 1 which is a considerable extrapolation of the model. Please see also our answer to further comment 6 of Referee #1 for why we can't report a percentage contribution of different surface types to the overall flux.

The key information we bring is to identify the relative important surfaces which have high emission potentials, which help to know which surfaces should be considered for conducting measurements.

*12. In table 2, you set up an investigation of scenarios to determine the level of complexity to use in the spatial disaggregation of fluxes. This is a great tool for supporting the robustness of your surface-type model results. You present results from the best model of the set described in the table. I would like to see more results on all scenarios. Specifically, how do the surface type flux estimates change in each version in table 2? In two of the five versions, your highest flux type is lumped with your lowest flux type, and discussing how the fluxes turn out in these scenarios would help provide confidence in the model results.*

**Answer:** In the revised  version of the manuscript we are reporting the estimated parameter values for full model with $\theta$ for 3,4,5 and 6 surface types in the supplement. In our opinion the results seem to provide more confidence that the ST estimates for the best model are coherent given the limited amount of data we have. We have added the following paragraphs to the results section on lines 535-545.

"Fig. S9-S12 show the estimated parameters for the full $\theta$ models for the other number of STs. Interestingly, for $CH_4$ when the two types of ditches are lumped into one ST, their $\gamma$ estimate is close to zero (Fig. S9 and S11) whereas when the ditches are considered as separate STs the estimated $\gamma$  for the plant covered ditch is the highest and the $\gamma$ for the ditches with water surface is the lowest which is the same behaviour what we see in Fig. 6 for the best model.

The parameter estimates between different number of STs for $N_2O$ models differ more than for $CH_4$ models. For example for ST6 (Fig. S12) the highest $\gamma$ MAP estimate is for dead wood and residue whereas the $\gamma$ for the field layer and trees is the smallest. The $\gamma$ estimates for ST5 (Fig.

S11) seem to also emphasize the role of litter and dead wood and residue as high $N_2O$ emitting surface types. It should be noted that for all other number of STs the living trees are always lumped together with some other surface type or types. It might be for this reason that the full $\theta$ no $\delta$ ST9 model outperforms the full $\theta$ ST6 model for $N_2O$ models but not for $CH_4$ models (Table S1)."

*Specific comments:*
*Line 88: Need space after period at the end of sentence*
*Line 107: Missing word. "[The] likely reason for this…*
*Line 247: missing space in citation for (Kljun et al 2015)*
*Line 565: Should cite Ludwig et al. 2024 here as well.*
*Line 581: Typo 'emissionsdd,*

**Answer:** These specific comments are included in the revised version of the manuscript as suggested.

---

## Author Response (AR2)

**Response to associate editor**

**Eddy covariance fluxes of CO2, CH4 and N2O on a drained peatland forest after clearcutting by Olli-Pekka Tikkasalo et al.**

We thank the associate editor for handling our manuscript and all the referees for their valuable comments to our work. Please find below the corrections made to the manuscript based on referee #2 and #3 comments.

**Referee #2**

*Thank you for your thoughtful responses to the reviewer questions. The manuscript is much improved. I am satisfied with all changes, and have only two minor points that might be addressed before publication. Figure 6 is still difficult to distinguish the bold line for inner quartile from the thin line for 95% HDI. Why not just use a boxplot instead of lines?*

**Answer:** We thank referee #2 again for reviewing our manuscript. We have increased the linewidth of the central quartiles such that they resemble a boxplot.

*Type line 550: "The predicted cumulative CH4 emission is an 60% smaller than that based " should be "The predicted cumulative CH4 emission is 60% smaller than that based"*

**Answer:** This typo has been corrected.

**Referee #3**

*This study presents valuable insights into the greenhouse gas fluxes (CH4, N2O, CO2) from a drained peatland forest following clearcutting. The study employs an innovative approach by combining eddy covariance (EC) measurements, statistical modeling, and surface-type classification using UAV imagery to understand the spatial and temporal variability of gas emissions.*

*Strengths of the Manuscript:*
*Limited data: There is currently limited EC data from boreal peatland clearcuts. This paper will provide timely information for decision making.*
*Novel Approach: The integration of surface-type classification with EC and statistical modeling is a notable strength of the manuscript. The use of UAV-based surface classification and Bayesian inference methods adds significant value to the methodology.*
*Detailed Data Analysis: The thorough analysis of the seasonal and spatial variability of CH4 and N2O emissions is comprehensive, and the modeling framework provides a solid foundation for understanding the drivers of these emissions in post-clearcut peatland ecosystems.*

**Answer:** We thank referee #3 for reviewing our manuscript and providing suggestions for improvement.

*Suggestions for Improvement:*
*Estimation of Uncertainty: One critical area that requires attention is the estimation of uncertainty in the results. This is particularly important when dealing with gases of small fluxes, such as CH4 and N2O. The study did not provide an analysis on the uncertainty, and this is crucial as there are multiple sources of uncertainty that could impact the results, such as model uncertainty, gap-filling uncertainty, and system uncertainty from the EC systems. A discussion or quantification of these uncertainties would help strengthen the robustness of your findings. I suggest to consider using Monte Carlo and provide a standard deviation to the flux estimates.*

**Answer:** We agree with the reviewer that uncertainty quantification is an important part of any scientific exercise. In this work, the uncertainty of annual GHG budgets derived from EC observations was estimated by gapfilling the flux time series with multiple gapfilling algorithms and by reporting the spread between the annual flux estimates. See Table 3 in the manuscript for the spread and description of the methodology in Sect. 2.3. We recognize that this approach misses some of the sources of uncertainty, e.g. random uncertainty of EC observations and uncertainties related to the post-processing of EC data, and hence likely slightly underestimates the total uncertainty. However, to our understanding standardized approaches for estimating uncertainties have not yet been developed, in particular for $CH_4$ and $N_2O$ annual emissions, and we argue that development of improved approaches is out of the scope of this manuscript. Hence, we opt to keep the uncertainty estimation as it currently is in the manuscript. However, we have added the uncertainty estimates to the manuscript abstract and added a short note on the shortcomings of this uncertainty estimation approach on line 214-217.

"However, it is possible that the spread may underestimate the total uncertainty of annual fluxes, since it does not take into account, for example, the contribution of random uncertainty associated with EC observations, and it relates only to uncertainty related to the gapfilling process."

In regards of modeling uncertainty, the 95 % confidence intervals for the modelled fluxes are already given in the Table 3. The confidence intervals were calculated based on the model parameter uncertainties.

*Spatial Partitioning of EC Data: The study rightly acknowledges the spatial nature of the EC source area; however, I believe more attention is needed to the challenges of applying EC data to narrow features like ditches, which contribute less than 2% of the EC footprint. While it is interesting to explore spatial partitioning and the variability across different surface types, it is important to note that such narrow features, like ditches, are often difficult to assess with EC systems. The fluxes from these areas are likely to be diluted by the surrounding landscape, which could result in an underestimation of the actual fluxes. While this is an interesting approach, it should be mentioned that the results may not fully capture the fluxes from these narrow features.*

**Answer:** We have added short note to discussion on lines 650-653 about the spatial representativeness of surface types in the clearcut and how that might transfer to inferred parameters

"Another possibility is that for some of the surface types their proportion inside the footprint is always so low that their contribution to the model estimate is diluted by the surrounding landscape (e.g., ditches with water surface). As a result the model might not correctly capture their contribution to the flux."

*Site Characterization: The manuscript lacks sufficient demonstration of site-specific conditions, particularly regarding the fertility and hydrological status of the study area. Information on these conditions is essential to contextualize the observed fluxes and to understand the underlying environmental drivers. Providing more detailed background on soil fertility and the hydrological conditions at the site would strengthen the manuscript and help interpret the results in a more meaningful way.*

**Answer:** We have added information about the fertility and hydrological status to the manuscript to the methods section (lines 143-145).

*Conclusion: Overall, this manuscript offers valuable insights into greenhouse gas emissions in boreal peatland forests post-clearcutting. The methodology is innovative and provides a strong foundation for further research. However, addressing the points outlined above— particularly the estimation of uncertainty, spatial partitioning challenges, and more detailed site characterization—would significantly improve the robustness of the manuscript.*

**Answer:** We thank referee #3 for reviewing our manuscript.

*1. Line 34: " Greenhouse gas (GHG) fluxes have been quantified (Ojanen et al., 2010)". The citation appears a bit strange to me as there should be many GHG related papers before and after Ojanen et al 2010.*

**Answer:** This sentence is related to the quantification of GHG fluxes in Finland. We have added additional citations that measure the GHG balance in Finland.

*2. Line 47: Rotation forestry (The term R is capitalized)*

**Answer:** We have changed the r to lowercase

*3. Line 51: What is the "duration" of GHG fluxes? Please explain or use a clearer term for that.*

**Answer:** We have changed here "GHG fluxes" to "GHG emissions"

*4. The paragraph #41-#56: I think you did a clear illustration on introducing rotation forestry. However, I think this paragraph could be shortened or even combined with other paragraphs. This manuscript does not study the entire rotation period, but just one year after clearcutting. Of course clearcutting is a part of rotation forestry but it should not be a main theme that*

*worths 16 lines to introduce. Specifically, the line regarding DOC and biodiversity of rotation forestry is irrelevant and so could be removed.*

**Answer:** We feel that this paragraph is crucial for outlining the importance of the study as it considers the short-term impacts of clearcutting. For this reason we shortened the paragraph only slightly.

*5. Line 126-129: Is there any reference to the climate data? Did you do the vegetation inventory and peat depth measurements yourselves?*

**Answer:** We have added the source for the climate data information. We measured the peat depth and made vegetation surveys.

*6. Line 139: "It was completed in June 2021 in the north-western section of the CC area" What is "it"? So the harvesting was primarily done 18-Mar to 1-Apr but was completed in June?*

**Answer:** We have changed "it" to "the harvests".

*7. Section 2.1: You mentioned that "the site is a fertile and well-drained", but provided no information about that. It would be beneficial to provide more information, including CN ratio, mean WTD prior to (if available) and after clear cut.*

**Answer:** We have added this information to the methods section as suggested.

*8. Line 155: You mentioned that the EC tower is 3.1 m tall. I assume that the canopy is about zero, then using the 1:100 rule of thumb, the flux footprint 90% should be about 300m. But based on figure 2, the footprint is less than 200m. Is there any reason to this?*

**Answer:** Like stated in the manuscript, the clearcut surface is a complex mosaic of vegetation patches and logging debris with underlying undulating surface. Hence, the displacement height caused by these flow obstacles is not zero and we estimated it empirically from the observations, see the manuscript for details. Due to this non-zero displacement height, the footprint is smaller than what one would simply estimate using the rule-of-thumb and measurement height.

*9. Line 226-228: So the N2O gapfilling model has the best performance among the three GHGs in terms of R2? I am actually quite surprised of that given the small magnitude of N2O fluxes and the complexity of the gas. In the same paragraph, you mentioned only the input variables for CH4 gapfilling but what variables did you use to gapfill N2O so its performance is so good?*

**Answer:** We agree that it is slightly surprising to see that the gapfilling model has the best performance for $N_2O$ fluxes. We assume that the good performance is since 1) $N_2O$ fluxes were easily detectable at this site (high signal-to-noise ratio) and 2) there is strong seasonality in $N_2O$ fluxes, and the emissions were not as sporadic and peaked as e.g. in agricultural sites. The gapfilling model was able to capture the seasonality accurately with the normalized daily incoming potential solar radiation and its first time derivative, i.e. it was able to explain bulk of

the variability. The list of predictors used to predict N2O emissions is already mentioned in the manuscript, on lines 225-229.

*10. Line 323-324: If I understand correctly, you developed the models separated for each land surface type? If so, then how did you derive the soil moisture? Did you use a single value for all land surface types, or did you consider the spatial variation of soil moisture (also soil temperature) as I assume the ditches can behave very differently ?*

**Answer:** The model parameters are fit simultaneously to each surface type using the proportion of each surface type inside the 30-min flux footprint, air temperature, soil moisture as model input and then the model estimate is compared against flux measured with the EC. Because the spatial coverage of the footprint changes, so does the fraction of each surface type in the corresponding flux value which the model estimate is compared against. This allows us to fit all the model parameters (including surface type specific parameters) simultaneously.

Unfortunately, we did not have spatially large enough sample for using different soil moisture parameter for each surface type. The soil moisture measurements were performed in the three locations shown in Fig. 1 and we use the mean of these three measurement points. Soil moisture indeed can vary quite a lot and for this reason we do not include surface type dependent soil moisture terms in the statistical models which estimate $CH_4$ and $N_2O$ flux.

*11. Line 407: Consider using a more updated GWP based on the recent IPCC reports.*

**Answer:** We have updated the GWP100 values from the latest IPCC report. The result affects significantly only $N_2O$ emission estimate.

*12. Figure 2A: Consider removing the green colour for NEE, and showing only as a line. It looks a bit misleading now, for instance, that there is only NEE in winter but no Reco.*

**Answer:** We have adjusted Fig. 2A as suggested.

*13. Figure 2D: You mentioned that you have 3 soil moisture monitoring points in Figure 1. You can consider showing the variation of Tsoil by shades also.*

**Answer:** We have added variation to the soil moisture plot.

*14. Figure 2: You have mentioned in the introduction that a fluctuating WTD could be a hot spot for N2O emissions. Have you considered also showing the time series of WTD?*

**Answer:** We opted not to show the WTD measurements because the WTD and soil moisture have a correlation of 0.72 i.e., the dynamics of WTD can be inferred from the dynamics of soil moisture. We have added the variation of WTD to the methods section as text.

*15. Figure 3: Did you include only measured flux data in the correlation analysis , or also the gapfilled data? If you include also the gapfilled flux data, do you think that this will interfere the correlation results?*

**Answer:** We included only the measured flux data except for GPP that needs the determination of $R_{eco}$ that is based on a statistical model. For this reason we cannot know how the non-gap filled GPP correlation would correlate with the other variables.

*16. Figure 3: Why only showing absolute value is higher than 0.25, instead of the statistically significant correlation? Also, it makes sense that P does not show any correlation with any variables if using 30-min data, but there should be some lag effects, so if you present also daily sum/means then the correlation result could be different. Indeed, correlation maps are symmetric so you can remove one side of the diagonal, and consider showing correlation result of daily means or other meaningful things you want to make use of the space.*

**Answer:** All the correlations presented are statistically significant. The choice of 0.25 is arbitrary. The main point of Fig. 3 is to show that the water availability related variables and temperature variables correlate with the flux values. Based on this correlation we build the models for determining surface type specific fluxes.

We have updated Fig. 3 and present now all the correlations in a lower triangle.

There might indeed be correlation if lag effects would be studied but we believe this is outside of the scope of this study.

*17. Figure 6: Is it that $\gamma$ denotes the strength of gas emissions, and $\delta$ denotes it dependence on the temperature? It seems that the information is not very clear neither in the figure or in the text.*

**Answer:** $\gamma$ doesn't directly translate to emission strength as there are also the soil moisture dependent term and the general $\alpha$ term in the model. Furthermore, the model predicts the natural logarithm of the flux which means that $\gamma$ from different surface are raised to the power of e and then multiplied together to calculate a flux value in units nmol $m^{-2}$ $s^{-1}$. For these reasons we decided not to give straightforward interpretation of the model parameters

*18. Figure 7: If $\gamma$ in figure 6 denotes the strength of gas emissions, then Figure 7 seems a bit repetitive. At least I see the relative difference across the land types are the same between $\gamma$ in figure 6 and figure 7.*

**Answer:** For the reasons outlined in above answer, we feel that Fig. 7 is needed as it shows how the model behaves if it would be used to extrapolate flux values from single surface type. The surface type specific flux is challenging to infer directly from Fig. 6.

*19. Line 526: "Finally, we calculated the total emissions for CH4 and N2O for the snow free period using the best models". I assume you mean gapfillig*

**Answer:** Here we mean the statistical models that calculate flux based on the surface type, air temperature and soil moisture. This sentence has been changed to

"Finally, we calculated the total emissions for $CH_4$ and $N_2O$ for the snow free period using the best full $\theta$ models"

*20. Line 630-631: If you separate the surface types in categories and calculate the $\delta$ and $\gamma$ separately for each surface type, then I do not think that adding C:N ratio would improve the model as it is just a constant through the year. But C:N ratio is definitely a very important number for you to explain the spatial variations.*

**Answer:** We agree with the referee and have clarified this sentence to

"Furthermore, the spatial variability of $N_2O$ emissions might be further explained by variables describing nutrient availability (e.g., C:N ratio)."

*21. Line 721: Please be consistent when writing the year in units (ie either a-1 or yr-1)*

**Answer:** We have changed the flux unit to $yr^{-1}$.